# Slug regulates the Dll4-Notch-VEGFR2 axis to control endothelial cell activation and angiogenesis

Nan W. Hultgren [1], Jennifer S. Fang [1], Mary E. Ziegler[1], Ricardo N. Ramirez [2], Duc T. T. Phan[1], Michaela M. S. Hatch[1], Katrina M. Welch-Reardon[1], Antonio E. Paniagua [3], Lin S. Kim[1], Nathan N. Shon [1], David S. Williams[3], Ali Mortazavi [2] & Christopher C. W. Hughes [1,4,5 ✉]

Slug (*SNAI2*), a member of the well-conserved Snail family of transcription factors, has multiple developmental roles, including in epithelial-to-mesenchymal transition (EMT). Here, we show that Slug is critical for the pathological angiogenesis needed to sustain tumor growth, and transiently necessary for normal developmental angiogenesis. We find that Slug upregulation in angiogenic endothelial cells (EC) regulates an EMT-like suite of target genes, and suppresses Dll4-Notch signaling thereby promoting VEGFR2 expression. Both EC-specific Slug re-expression and reduced Notch signaling, either by γ-secretase inhibition or loss of Dll4, rescue retinal angiogenesis in SlugKO mice. Conversely, inhibition of VEGF signaling prevents excessive angiogenic sprouting of Slug overexpressing EC. Finally, endothelial Slug (but not Snail) is activated by the pro-angiogenic factor SDF1α via its canonical receptor CXCR4 and the MAP kinase ERK5. Altogether, our data support a critical role for Slug in determining the angiogenic response during development and disease.

---

[1] The Department of Molecular Biology and Biochemistry, University of California Irvine, Irvine, CA 92697, USA. [2] The Department of Developmental and Cell Biology, University of California Irvine, Irvine, CA 92697, USA. [3] The Department of Ophthalmology, University of California, Los Angeles, Los Angeles, CA 90095, USA. [4] The Department of Biomedical Engineering, University of California Irvine, Irvine, CA 92697, USA. [5] The Edwards Lifesciences Center for Advanced Cardiovascular Technology, University of California Irvine, Irvine, CA 92697, USA. ✉email: cchughes@uci.edu

Blood vessel formation, which includes both vasculogenesis and angiogenesis, is a complicated multi-step process important for organismal development and wound healing[1,2]. Out of context, however, these processes can both trigger or exacerbate an array of human diseases, including retinopathy and cancer[3,4]. While recent advances have improved our understanding of the molecular processes underlying blood vessel growth, most therapeutic approaches to limit angiogenesis still focus on the VEGF pathway[3–5]. Yet, angiogenesis can proceed via alternative pathways—such as TGFβ/BMP, Wnt, Hedgehog, and Notch[6–9]—even when VEGF is limited[2,10,11].

We and others have previously noted similarities between angiogenic sprouting and epithelial-to-mesenchymal transition (EMT)[12,13], a term describing the process whereby polarized epithelial cells lose apical-basal polarity, adopt a mesenchymal phenotype, and migrate away from the parent tissue, either singly or in sheets[14,15]. In epithelial cells, EMT is driven by Snail family transcription factors, including Snail (SNAI1) and Slug (SNAI2)[14–16]. Endothelial tip cells that lead new sprouts can be viewed as having undergone a partial endothelial-to-mesenchymal transition (EndoMT), as they also lose apical-basal polarity, disengage from the endothelium of stable vessels, and lead the invading sprout[12,13], although they retain anterior-posterior polarity and junctional connections to the trailing trunk cells[13]. Further, endothelial cells (EC) are capable of undergoing full EndoMT under some conditions to become fibroblastic[12,17,18]. Previously, we found that both Slug and Snail are upregulated in EC in an in vitro angiogenesis assay, and that both are required for proper sprouting[19]. Interestingly, Slug overexpressing EC do not outcompete wild-type cells for the tip position, suggesting that Slug marks activated EC but is not alone sufficient for tip cell specification. We also showed that Slug regulates the matrix metalloprotease, MT1-MMP, but in that study identified neither upstream activators nor its downstream targets.

Slug and other Snail family transcription factors are critical for EMT, which is important in both development and pathological conditions, such as cancer metastasis and tissue fibrosis[16,20,21]. In mammals, Slug is expressed in numerous embryonic tissues; yet, the global Slug knockout (SlugKO) mouse is surprisingly viable though runted[22,23]. Studies in this mouse and other models show Slug regulates melanocyte stem cells, hematopoietic stem cells, and germ cells in mice[24] and neural crest-derived cells in human[25,26], but vascular defects have not thus far been noted. Interestingly, Slug does affect the cardiovascular system as overexpression results in cardiac defects during development[27], suggesting that further study is warranted.

Here we look to reconcile our previous in vitro data with the mouse knockout studies by performing a more detailed analysis of angiogenic vasculature (under both developmental and pathological conditions) in SlugKO mice, and complement this work with mechanistic studies using in vitro human angiogenesis models. In mice lacking Slug, we find that retinal angiogenesis is delayed and that, remarkably, growth of wild-type tumors is almost completely suppressed when invading host blood vessels lack Slug. We identify SDF1α as an upstream regulator of Slug expression in EC and identify Dll4-Notch signaling as a critical downstream effector of Slug during angiogenesis.

## Results

### Endothelial Slug regulates tumor angiogenesis and growth. To test whether Slug is important for pathological angiogenesis, we used a mouse syngeneic tumor model. CMT-93 mouse colorectal tumor cells[28] were injected subcutaneously into the flank of either wild-type (WT) or SlugKO mice. Tumor volume was measured daily for 6 weeks. While the tumors grew well in the WT mice, we

observed a significant decrease in tumor growth in SlugKO animals at all time points (Fig. 1a). We observed similar results with a second syngeneic colorectal tumor cell line, MC-38[29] (Fig. 1b and Supplementary Fig. 1a), as well as with a B16 melanoma cell line[30] (Supplementary Fig. 1b). Of note, in many SlugKO animals, tumor development was completely halted after a brief initial growth period, and some tumors even regressed after week 2 (Fig. 1a, b and Supplementary Fig. 1a, b). We confirmed by western blot that all three tumor lines express detectable amounts of Slug (Supplementary Fig. 2), as previously reported[28,29,31]. Thus, these data strongly suggest that loss of Slug in the tumor stroma—i.e. in tumor blood vessels, which originate from host tissue rather than the cancer syngraft—limits tumor progression. Specifically, we hypothesize that the absence of Slug in EC inhibited tumor angiogenesis.

To gain a clearer picture, we employed the Matrigel plug in vivo angiogenesis assay[32], and assessed resulting blood vessels in 7 days. Strikingly, while plugs from the WT mice were filled with blood, plugs from the SlugKO animals were relatively pale (Fig. 1c). Immunohistochemical staining confirmed significantly reduced CD31+ EC infiltration in the SlugKO plugs (Fig. 1d), measured as reduced percentage in CD31+ area (Fig. 1e), as well as a greater reduction in the percent area occupied by lumenized vessels in the SlugKO plugs (Fig. 1f), suggesting failure to form functional vessels.

Interestingly, we also observed reduced infiltration of CD31− stromal cells in SlugKO mice (Fig. 1d). Stromal cells, such as fibroblasts, are required during angiogenic sprouting to synthesize and remodel extracellular matrix[33]. To rule out whether our findings can be explained by aberrant recruitment and function of CD31− stromal cells, we conducted a rescue experiment using syngeneic WT mouse stromal fibroblasts (mF)[34]. Matrigel containing WT mF was unilaterally injected into the flank of SlugKO mice, with the other flank receiving a control injection of acellular Matrigel plug. We also implanted acellular plugs in WT mice as an additional control. Again, there was little vascular ingrowth in plugs from SlugKO mice compared to WT mice (Fig. 1g, h). Importantly, the inclusion of Slug-expressing mF failed to rescue vessel ingrowth in SlugKO mice (Fig. 1g, h), supporting our hypothesis that the loss of endothelial Slug accounts for the poor angiogenic response.

To directly assess tumor angiogenesis and its effect on tumor growth, we conducted immunohistochemical analysis on CMT-93 tumors from WT and SlugKO mice that had tumors large enough to harvest (Supplementary Fig. 1b). We observed significantly less vasculature in tumors from SlugKO mice, particularly in the center of the tumors (Supplementary Fig. 3a, b), as well as an increase in necrotic core area (Supplementary Fig. 3a, c). There was no reduction in tumor proliferation (via Ki67 staining) in SlugKO mice (Supplementary Fig. 3d, e). Together, these data indicate that the suppressed tumor development and impaired (pathological) tumor angiogenesis in SlugKO mice is likely due to EC-intrinsic defects of Slug deficiency.

### Slug regulates developmental angiogenesis in vivo. Mice are born with avascular retinas[35], thus although SlugKO mice appear to have intact retinal tissue development during embryogenesis[22], the impact of Slug deletion may still be evident in postnatal retinal vascular development. To test this, we analyzed neonatal retinas at multiple developmental stages. We first observed irregular and underdeveloped primary vascular plexus in SlugKO mice at postnatal day 5 (P5; Fig. 2a). Radial expansion of SlugKO vascular networks was delayed, as evidenced by reduced vessel length compared to WT (Fig. 2a, b). This is consistent with our previously published data showing Slug knockdown impairs

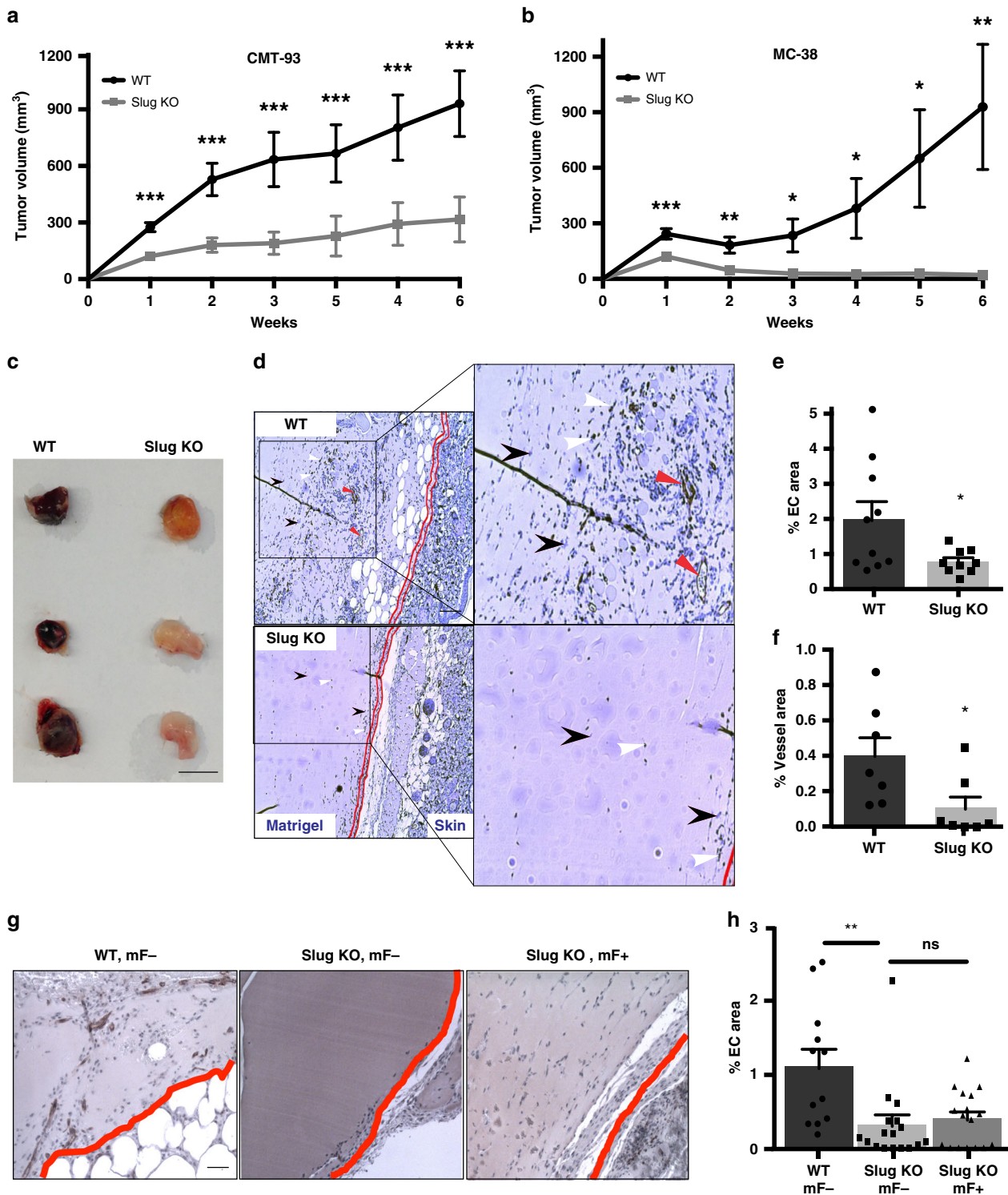

**Fig. 1 Slug deficiency inhibits tumor growth by reducing tumor angiogenesis. a** Growth curve of CMT-93 colorectal tumors in WT ($n = 24$) and SlugKO ($n = 27$) mice. **b** Growth curve of MC-38 colon tumors in WT ($n = 11$) and SlugKO ($n = 12$) mice. **c** Matrigel plugs harvested from WT and SlugKO mice. Scale bar: 1 cm. **d** CD31 staining of Matrigel sections reveals reduced EC recruitment (CD31$^+$ EC indicated by white arrow heads), reduced vessel formation (indicated by lack of lumenized vessels, shown by red arrow heads in control), and reduced stromal cellularity (indicated by decreased CD31$^-$/hematoxylin$^+$ cells, shown by black arrow heads) in Matrigel plugs of SlugKO mice. Double red lines indicate border between native mouse skin and Matrigel. Scale bar: 100 μm. **e** Average percent CD31$^+$ area in Matrigel sections from WT ($n = 10$) and SlugKO ($n = 9$) mice. $p = 0.036$. **f** Average percent lumenized vessel area in Matrigel sections from WT ($n = 7$) and SlugKO ($n = 7$) mice. $p = 0.037$. **g** CD31 staining of Matrigel sections indicates that the addition of WT mF cannot rescue the impaired EC recruitment in SlugKO mice. Red line indicates border between native mouse skin and Matrigel. Scale bar: 100 μm. **h** Average percent CD31$^+$ area in Matrigel sections from WT ($n = 12$) and SlugKO (mF−: $n = 17$, mF + : $n = 17$) mice. $p = 0.004$. Data represent mean ± SEM. Two-tailed unpaired equal-variance $t$ test. *$p < 0.05$, **$p < 0.01$, ***$p < 0.005$. Source data are provided as a Source Data file.

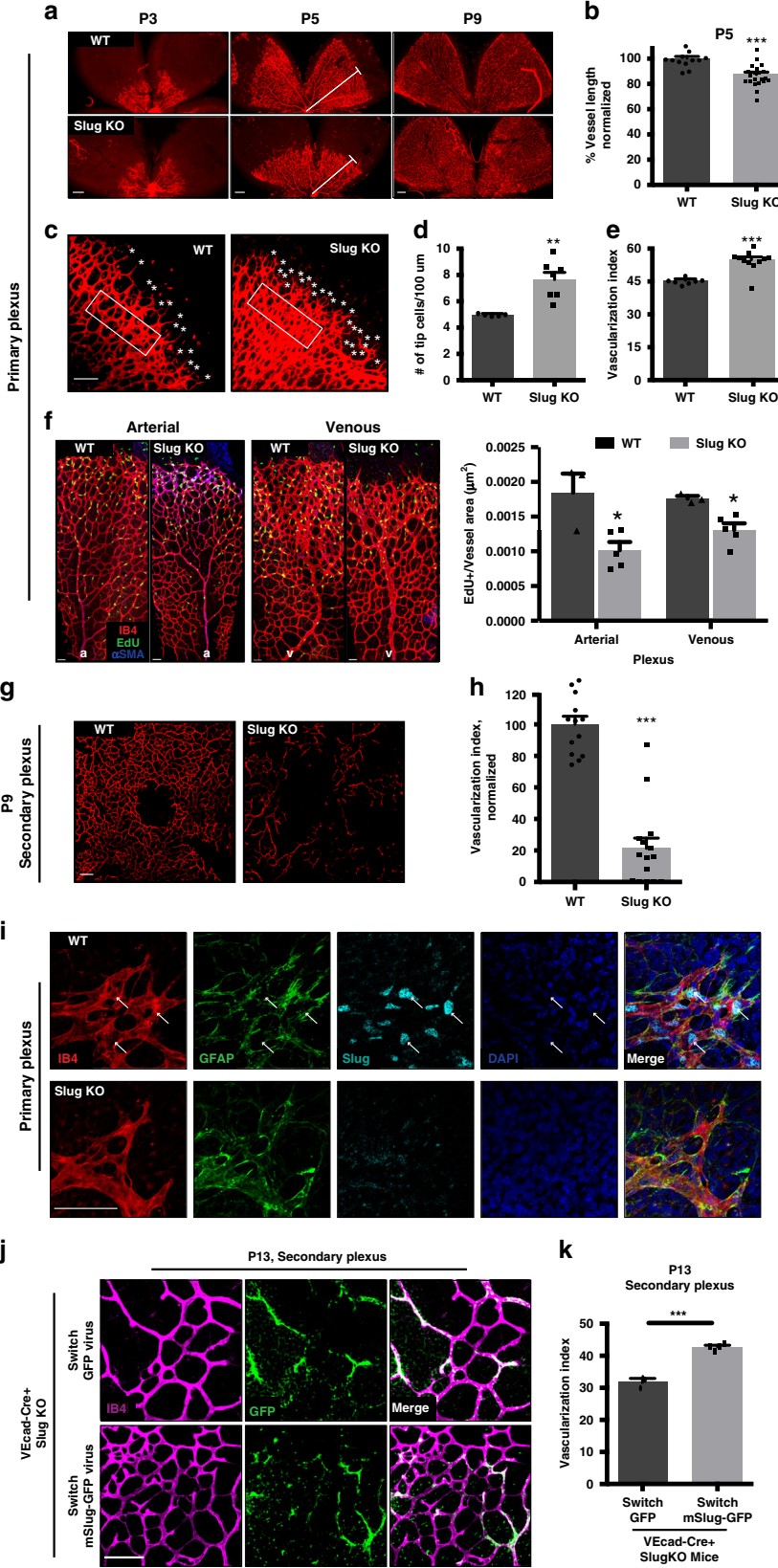

human EC migration in vitro[19]. In SlugKO mice, however, this defect was transient, and became less obvious by postnatal day 9 (P9; Fig. 2a and Supplementary Fig. 4a). Further examination of P5 SlugKO retinas revealed an increase in both tip cell number (Fig. 2c, d) and vessel density, quantified as vascularization index (Fig. 2c, e) at the vascular front. We also assessed the extent of vascular smooth muscle cell investment, visualized by the mural cell marker αSMA, as a measure of arterialization, and found no difference between WT and SlugKO tissues (Supplementary Fig. 4b).

**Fig. 2 Slug deficiency leads to impaired neonatal retinal vascular development. a** IB4 staining of WT and SlugKO mouse retinal primary plexus at P3, P5, and P9. Scale bar: 100 μm. **b** Average vessel length as a percentage of the retinal leaflet is quantified for P5 WT ($n = 12$) and SlugKO ($n = 21$) mice and normalized to WT as 100%. $p = 0.0001$. White line in the middle panel in **a** indicates vessel length measured. **c** IB4 staining of P5 WT and SlugKO mouse retinal primary plexus vascular front. White rectangle specifies the area analyzed for leading edge vascularization index (percentage of retinal area occupied by vessels). White star specifies tip cells. Scale bar: 100 μm. **d** Average number of tip cells per 100 μm in WT ($n = 5$) and SlugKO ($n = 7$) mice. $p = 0.002$. **e** Average vascularization index at the leading edge of primary retinal vasculature in WT ($n = 7$) and SlugKO ($n = 11$) mice. $p = 0.0001$. **f** Left: IB4, αSMA, and EdU staining of P6 WT and SlugKO mouse retinal primary plexus. Right: average number of EdU$^+$ cells per IB4$^+$ vessel area (μm$^2$) in arterial ($p = 0.02$) and venous ($p = 0.01$) regions in WT ($n = 3$) and SlugKO ($n = 5$) retinal vessels. Scale bar: 50 μm. **g** IB4 staining of WT and SlugKO mouse retinal secondary plexus at P9. Scale bar: 100 μm. **h** Average vascularization index in the secondary plexus of the entire retina quantified for P9 WT ($n = 13$) and SlugKO ($n = 15$) mice, normalized to WT as 100%. $p < 0.0001$. **i** IB4, GFAP, Slug, and DAPI staining of WT and SlugKO mouse retinal primary plexus at P5. Scale bar: 100 μm. **j** IB4 staining and GFP expression in VEcad-Cre$^+$, SlugKO mice injected with Switch-GFP or Switch-mSlug-GFP virus. Scale bar: 100 μm. **k** Average vascularization index in the secondary plexus at P13 is quantified for VEcad-Cre$^+$, SlugKO mice injected with Switch-GFP ($n = 3$) or Switch-mSlug-GFP ($n = 4$) virus. Vascularization is increased when Slug expression is restored to EC. Scale bar: 100 μm. $p = 0.0001$. Data represent mean ± SEM. Two-tailed unpaired equal-variance $t$-test. *$p < 0.05$, **$p < 0.01$, ***$p < 0.005$. Source data are provided as a Source Data file.

To determine if the vascularization delay is due to changes in retinal EC proliferation, we performed EdU labeling in vivo in P6 WT and SlugKO mice. As in other cell types[36], we found that Slug deficiency decreases the number of EdU$^+$ EC in both arterial and venous areas (Fig. 2f). Thus, delayed vascular expansion is likely the result of reduced EC migration (leading to the accumulation of EC at the vascular front), combined with reduced EC proliferation.

Next, we examined the deeper vascular layer. As expected, the vascular density at the secondary plexus of SlugKO mice was severely reduced by ~80% at P9 relative to littermates (Fig. 2g, h). Again, this defect was transient: no difference was observed between WT and SlugKO mice by P15 (Supplementary Fig. 4c).

To assess whether embryonic blood vessel beds are also impacted by Slug deletion, we examined mouse yolk sac vasculature at E8.5. Compared to WT, SlugKO yolk sac vessels appeared more disorganized, with an underdeveloped hierarchy, fewer branches and increased vessel diameter (Supplementary Fig. 4d).

**Slug expression in retina is restricted to blood vessels.** Retinal vascular development is dependent on the cooperation of multiple cell types, including astrocytes and pericytes. To distinguish which cell types might contribute to the vascular defects in SlugKO mice, we examined Slug expression in vascular and perivascular cells of the neonatal mouse retina. As expected, we observed EC nuclear Slug expression throughout WT vessel networks including in tip cells, whereas Slug signal was not evident in SlugKO retinal EC (Fig. 2i and Supplementary Fig. 5a, b). By contrast, we did not find Slug$^+$ astrocytes (Fig. 2i and Supplementary Fig. 5b), nor was there an obvious change in astrocyte (GFAP$^+$) distribution in SlugKO mice (Supplementary Fig. 5c). Thus, vascular defects in SlugKO mice are likely not due to the absence of Slug in retinal astrocytes. Although we found Slug expression in pericytes (NG2$^+$), especially along arteries (Supplementary Fig. 5d), there was no significant difference in pericyte distribution and coverage between WT and SlugKO mice (Supplementary Fig. 5e). These data emphasize the important role that Slug plays in EC but not astrocytes or pericytes during retinal vessel development.

Snail (*Snai1*) reportedly plays a compensatory role in cardiac valve formation[22] and chondrogenesis[37] in SlugKO mice. We therefore examined whether Snail might also compensate for Slug deletion in mouse retinal EC. We used fluorescence-activated cell sorting (FACS) to isolate CD31$^+$ EC from WT, Slug heterozygous (Slug Het), and SlugKO retinas, and assessed gene expression of both *Snai1* and *Snai2*. We found no significant difference in *Snai1* expression between the three genotypes (Supplementary Fig. 5f), suggesting that increased *Snai1* is likely not compensatory in this context.

To further confirm the importance of endothelial Slug in driving the vascular phenotype in SlugKO retinas, we performed an EC-specific rescue of Slug expression in vivo. We first generated a lentiviral Switch construct, wherein a dsRed-STOP expression cassette is flanked by two LoxP sites and followed by a full-length mouse Slug-GFP expression cassette (Supplementary Fig. 6a). When transduced, Cre$^-$ cells express only dsRed, whereas Cre$^+$ cells excise the stop codon to enable mSlug (and GFP) expression. We generated both control Switch-GFP and Switch-mSlug-GFP virus and tested each in vitro by transducing a mixed Cre$^+$ and Cre$^-$ EC population. We found distinct dsRed and GFP expressing populations in both control Switch-GFP and Switch-mSlug-GFP transduced EC as expected (Supplementary Fig. 6b). However, only the Switch-mSlug-GFP transduced EC showed Slug overexpression (Supplementary Fig. 6c). To achieve EC-specific Slug re-expression in vivo, we crossed SlugKO mice with mice that selectively express Cre in vessel endothelium (via EC-specific VE-cadherin promoter, VEcad-Cre). When Switch-GFP or Switch-mSlug-GFP virus was injected retro-orbitally into VEcad-Cre$^+$ SlugKO mice, GFP expression was limited to EC (Fig. 2j). For technical reasons we injected mice at P6 and allowed 1 week for viral uptake and gene re-expression. By this time (P13), vascular defects in SlugKO mice were most obvious in the secondary (rather than primary) plexus (Supplementary Fig. 4a). VEcad-Cre$^+$ SlugKO mice that received Switch-mSlug-GFP virus showed increased vascularization of the secondary plexus compared to those that received control Switch-GFP virus (Fig. 2J, K). Importantly, VEcad-Cre$^-$ SlugKO mice that received Switch-mSlug-GFP virus showed no change in vascularization (Supplementary Fig. 6d, e). Injected mice showed no obvious vascular injury related to the procedure itself, as compared to control mice with no injection (Supplementary Fig. 6d, e).

Together, these studies indicate that in addition to its role in pathologic angiogenesis, EC-expressed Slug is also important in developmental angiogenesis.

**Slug expression in EC regulates vessel formation in vitro.** To determine if Slug affects vascular morphogenesis in a dose-dependent manner, we overexpressed Slug at both a low (SlugOE$^{low}$) and high (SlugOE$^{high}$) level (Supplementary Fig. 7a), with the SlugOE$^{low}$ levels similar to those induced during sprouting angiogenesis[19]. During the early phase of sprouting angiogenesis, both SlugOE$^{low}$ and SlugOE$^{high}$ promoted—in a dose-dependent manner—tip cell formation and an earlier appearance of sprouts (Fig. 3a; lower magnification, Supplementary Fig. 7b). However, during the later stage, both SlugOE$^{low}$ and SlugOE$^{high}$ disrupted vessel maturation and lumen formation. Specifically, while SlugOE$^{low}$ caused enlarged lumens,

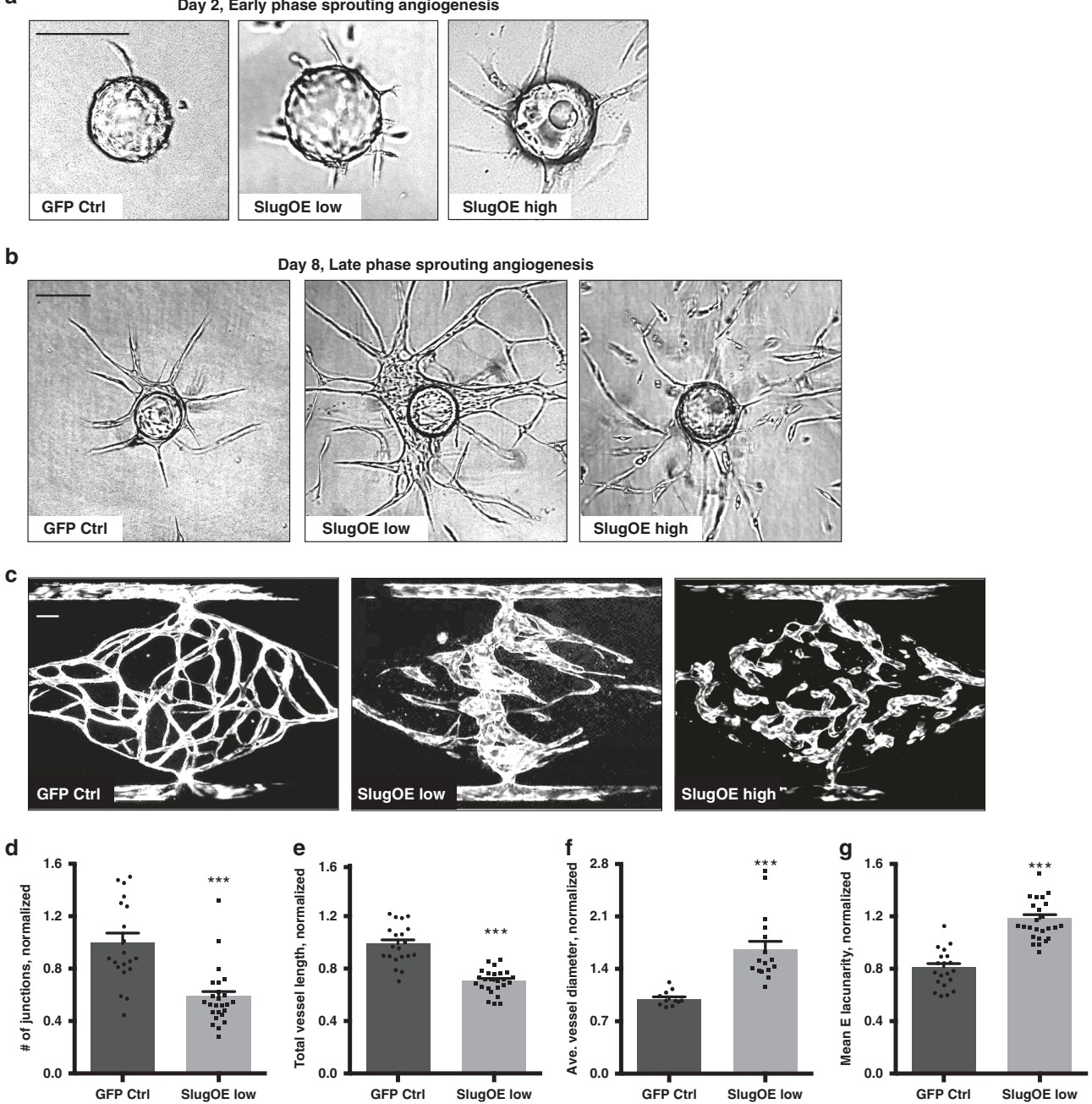

**Fig. 3 Slug overexpression in EC disrupts vascular morphogenesis. a** SlugOE in EC induces increased tip cell formation (middle) and accelerates sprouting (right) in a dose-dependent manner during early phase sprouting angiogenesis in the fibrin-gel bead assay. Scale bar: 100 μm. **b** SlugOE at low levels in EC leads to dilated lumens (middle), and fragmentation of the sprouts at high levels (right) in the fibrin-gel bead assay. Scale bar: 100 μm. **c** SlugOE at low levels in EC leads to dilated lumens (middle), and fragmentation of the network at high levels (right) in the in vitro vascularized micro-organ model of vasculogenesis. Scale bar: 100 μm. This is quantified as **d** reduced number of junctions (GFP $n = 20$ devices, SlugOE $n = 25$ devices. $p < 0.0001$), **e** reduced total vessel length (GFP $n = 20$ devices, SlugOE $n = 25$ devices. $p < 0.0001$), **f** increased average vessel diameter (GFP $n = 11$ devices, SlugOE $n = 16$ devices. $p < 0.0001$), and **g** increased average lacunarity (GFP $n = 20$ devices, SlugOE $n = 25$ devices. $p < 0.0001$). Data represent mean ± SEM. Two-tailed unpaired equal-variance $t$-test. $*p < 0.05$, $**p < 0.01$, $***p < 0.005$. Source data are provided as a Source Data file.

SlugOE[high] led to disconnected sprouts and even single cells (Fig. 3b; lower magnification, Supplementary Fig. 7c), reminiscent of an EMT.

To determine whether SlugOE in EC also affects vasculogenesis, we used an in vitro vascularized micro-organ (VMO) platform[38] in which EC form a capillary bed that anastomoses to microfluidic "arteries" and "veins" allowing flow of a blood substitute through the network. Compared to control EC, SlugOE[low] cells formed distorted vessels with larger diameters and fewer branch points (Fig. 3c–g), while SlugOE[high] cells were mostly incapable of forming networks (Fig. 3c). In addition, EdU staining revealed increased proliferation in SlugOE[low] EC (Supplementary Fig. 7d), consistent with our results in the retinal angiogenesis model (Fig. 2h), and previous reports[36]. In aggregate, these data demonstrate that Slug regulates blood vessel formation in a dose-dependent manner.

**RNA-seq reveals an activation profile in SlugOE EC**. To explore the underlying mechanism by which EC-expressed Slug regulates vascular formation, we performed RNA-seq on EC harvested from the fibrin-gel bead assay. We compared the gene expression profiles between GFP-control and GFP-SlugOE EC at day two (D2) and six (D6) of the assay, which represents the initiation of sprouting and the establishment of lumenized vessels, respectively. EC from three different isolations were used to minimize the effect of donor variability (Supplementary Fig. 8a), and each was processed separately. We identified 1992 differentially expressed genes (Fig. 4a), and performed K-means clustering, which recognized 10 clusters with distinct expression profiles (Fig. 4a and Supplementary Data 1). Gene ontology (GO) analysis was performed on genes from each cluster, which identified a cluster-specific enrichment pattern, including terms related to cell proliferation, migration, junction assembly, and matrix protein regulation, as well as cytoskeleton regulation and cell shape (Fig. 4b). The top two ranked GO terms from each cluster were plotted (Fig. 4c).

Several clusters were of particular interest. Both Clusters 1 and 3 contain many downregulated junctional genes, while cluster 5 shows strong upregulation of genes in proliferation and migration at day 6. Proliferation genes are also strongly associated with cluster 7, but only at day 2. Thus, cluster 7 may represent genes involved in proliferation during early stage of angiogenesis, while cluster 5 is presumably associated with genes controlling proliferation at later times.

Many EMT signature genes were differentially regulated during in vitro sprouting angiogenesis and by Slug overexpression (Supplementary Fig. 8b). EMT was also among the top-ranked GO terms in clusters 3 and 5 (Fig. 4c). We confirmed the increased expression of several EMT-related genes, including *ACTA2*, *SPARC*, *CCND1*, and *COL1A1* in SlugOE EC at both RNA (Supplementary Fig. 8c) and protein level (Supplementary Fig. 8d). EMT genes are found in both up- and downregulated clusters (Fig. 4c, Supplementary Fig. 8b). Downregulated EMT genes include both junctional genes and known negative regulators of EMT such as *FOXP4* and *HHIP* (Supplementary Data 1).

Gene Set Enrichment Analysis (GSEA) indicates that SlugOE leads to upregulation of genes enriched in the TGFβ receptor pathway, regulation of cell cycle, shape and cell movement, whereas genes regulating cell junctions and maturation are downregulated (Fig. 4d and Supplementary Fig. 8e, f).

Since the loss of junctions is a hallmark of an EMT/EndoMT event[16,39], we focused on this finding. To verify changes at the protein level, we performed immunofluorescence staining and found a reduction in the surface expression of Claudin5 and VE-cadherin (Fig. 4e). Interestingly, not all junction proteins were downregulated at the transcriptional level, notably *CLDN5* (Supplementary Fig. 8f and Supplementary Data 1). To test if the global reduction of junction proteins impacted EC function, we analyzed vascular leakage in the VMO model. Although SlugOE EC still self-assemble into vessels—suggesting that junctional complexes still form despite reduced expression of some components—we observed greater leakage of luminally-perfused 70 kDa fluorescent dextran in networks comprised of SlugOE EC vs. control (Fig. 4f). To assess if vascular permeability is similarly affected in vivo, we performed the Miles Assay and found that SlugKO mice are less prone to VEGF-induced vascular hyperpermeability than are WT mice (Supplementary Fig. 9).

Taken together, these data support the involvement of endothelial Slug in driving the transition of EC towards a mesenchymal-like state—a process we call partial EndoMT, as EC both retain endothelial characteristics and do not fully detach from adjacent cells.

**Slug regulates Dll4-Notch-VEGFR2 axis in angiogenic EC**. Notch signaling is among the most important pathways involved in vascular development[6,40–42]. Our RNA-seq study identified changes in gene expression for several components of the Notch pathway, including reduced expression of Notch ligand *DLL4*[41], and EC-specific Notch target *HEY1*[43] (Fig. 5a). To validate these results, we performed siRNA-mediated Slug knockdown in SlugOE cells and observed strong induction of *DLL4* and *HEY1* at both RNA (Fig. 5b) and protein level (Fig. 5c and Supplementary Fig. 10b). Similar results were observed with siRNA-mediated knockdown of endogenous Slug using a second siRNA (Supplementary Fig. 10a). Importantly, we observed reduced levels of cleaved Notch intracellular domain (NICD) in SlugOE EC (Fig. 5c and Supplementary Fig. 10b), confirming that Slug decreases Notch activation in EC.

Slug modulates transcription by binding to E-box sequences in the regulatory regions of its target genes[16]. There are several E-box sequences (CANNTG) proximal to the *DLL4* transcription start site (TSS) in both mouse and human[44,45], albeit in different locations. To test direct binding of Slug to the *DLL4* promoter, we performed chromatin immunoprecipitation (ChIP)-qPCR with primers flanking the two E-box sequences at +157/+162 and +178/+183 of the human *DLL4* gene (Fig. 5d). Compared to the control gene *ALB*, we observed a significant enrichment for *DLL4* E-box binding sites in SlugOE EC (Fig. 5d and Supplementary Fig. 10c). Slug also regulated a second Notch ligand, *JAG2* (Fig. 5a), likely also by direct binding to its promoter (Supplementary Fig. 10d). Therefore, Slug is capable of directly binding to the promoter region of Notch ligands and likely acts as a transcriptional repressor.

Consistent with previous reports[43,46–48], we found reduced *KDR* (VEGFR2) expression downstream of increased *DLL4* and Notch activation in Slug knockdown EC (Fig. 5b). Conversely, SlugOE leads to decreased Notch activation and increased VEGFR2 (Fig. 5c). To determine whether the hypersprouting phenotype in SlugOE cells might be due to high VEGFR2, we used the fibrin-gel bead assay and titrated in VEGFR2 inhibitors Apatinib and Pazopanib. Both VEGFR2 inhibitors curbed the Slug-driven hypersprouting phenotype in a dose-dependent manner (Fig. 5e). Further, the number of disconnected fragments also decreased (Fig. 5f), suggesting that reduced VEGF signaling also attenuated EndoMT in SlugOE EC. To rule out off-target effects, we also titrated down VEGF concentration and, again, completely corrected the hypersprouting and EndoMT phenotype in SlugOE EC (Fig. 5g, h).

To test whether direct Notch inhibition synergizes with SlugOE, we treated either control or SlugOE EC with the Notch inhibitor, DAPT, which blocks γ-secretase activity. While DAPT induced hypersprouting in control cells[41], treatment in SlugOE cells resulted in complete disassembly of sprouts into single cells (Supplementary Fig. 11a, b). Together, these data confirm that endothelial-expressed Slug promotes sprouting angiogenesis by modulating Notch signaling via transcriptional repression of *DLL4* and upregulation of *KDR*.

**Notch inhibition rescues retinal angiogenesis in SlugKO mice**. Our in vitro data suggested that Notch inhibition might rescue vascular defects in vivo in SlugKO retinas. Consistent with previous studies[49], Dll4 is expressed in WT retinal EC at the leading vascular front (Fig. 6a). However, in SlugKO animals, we observed mis-expression of Dll4, with signal present both at the leading edge and in the middle of the developing vessel network (Fig. 6a). Moreover, VEGFR2 expression was significantly reduced in areas expressing Dll4, particularly in SlugKO retinal EC (Fig. 6b).

Next, we treated WT and SlugKO mice with DAPT for 48 h to inhibit Notch signaling. DAPT treatment significantly increased

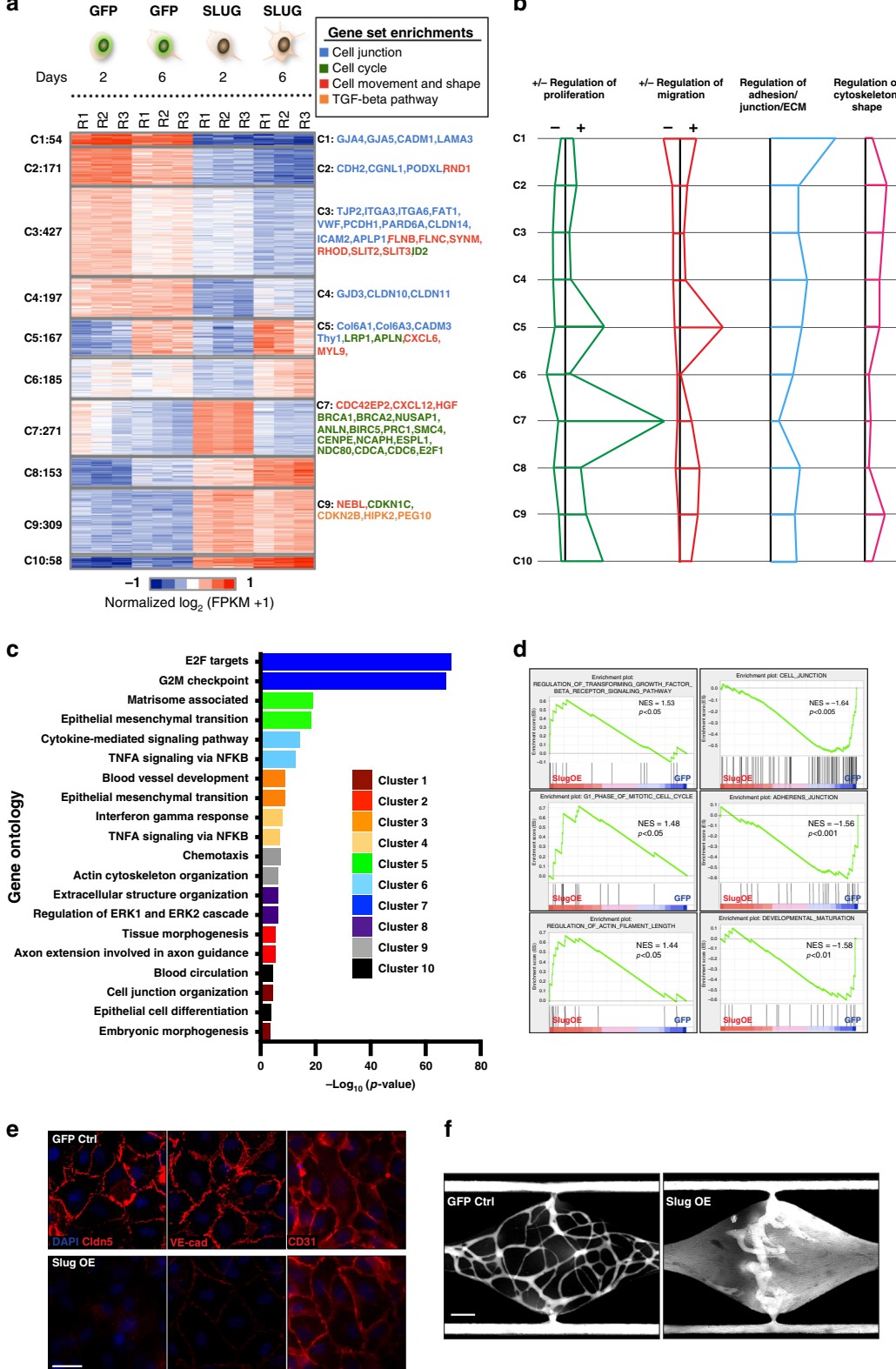

vascular expansion in P5 SlugKO retinas, restoring outgrowth to that observed in WT littermates (Fig. 6c). Therefore, Notch inhibition rescues delayed retinal angiogenesis in SlugKO mice.

Finally, to confirm that Slug regulates retinal angiogenesis by modulating Dll4 expression, we crossed SlugKO mice with Dll4 heterozygous (Dll4 Het) animals. Consistent with our DAPT study, Dll4 haploinsufficiency on a SlugKO background (SlugKO/

Dll4Het) almost completely restored the delayed retinal vascular expansion in SlugKO mice (Fig. 6d).

**SDF1α-CXCR4-ERK5 induces angiogenesis by activating Slug.** SDF1α is a well-known pro-angiogenic factor[50], and its expression is elevated in tumors[51,52]. SDF1α treatment (100 ng/mL)

**Fig. 4 RNA-seq analysis reveals an EndoMT gene expression profile. a** Left: Heat map for differentially expressed genes in GFP control and SlugOE EC on day 2 and day 6 of the fibrin-gel bead assay, grouped into 10 unique K-mean clusters. Right: Representative genes from each of the four main GSEA terms as shown in their respective K-mean clusters. **b** Differences in enrichment of indicated GO terms across 10 K-mean clusters. Length of the horizontal line for each cluster indicates the proportion of genes associated with that GO term that are enriched or reduced in the cluster. When present, + and − indicates the direction (positive and negative, respectively) of regulation for each biological process (Proliferation and Migration). Direction of regulation is not specified for Adhesion/Junction/ECM and Cytoskeleton/Shape. **c** Top two ranked Gene Ontology terms enriched in each K-mean group. **d** GSEA analysis showing enrichment in EMT-related terms in the gene set upregulated in SlugOE EC and downregulated in GFP control EC. **e** Immunofluorescence staining showing reduced surface expression of Claudin5 and VE-cad, but not CD31 in SlugOE EC. Scale bar: 50 μm. **f** 70 kDa rhodamine dextran perfusion of the vascularized micro-organ model at day 5 showing that network formed by SlugOE EC is leaky. Scale bar: 200 μm.

increased tip cell formation in the fibrin-gel bead assay (Fig. 7a and Supplementary Fig. 12). Concurrently, *SNAI2*, but not *SNAI1* transcription was increased (Fig. 7b). In a mouse retinal explant model, SDF1α treatment led to increased sprouting in WT, but not in SlugKO explants (Fig. 7c, d), indicating that SDF1α promotes angiogenesis via Slug.

SDF1α has two known receptors in EC, CXCR4 (*CXCR4*) and CXCR7 (*ACKR3*)[51]. Although CXCR4 is thought to be the canonical receptor, some recent studies have suggested a role for CXCR7 in tumor angiogenesis[53,54]. To determine which receptor is used by SDF1α to induce Slug, we knocked down each of them via siRNA in EC (Supplementary Fig. 13a) and checked for Slug induction upon SDF1α treatment. Consistent with previous findings[52,55], knockdown of either CXCR4 or CXCR7 blocked sprouting in the fibrin-gel bead assay (Supplementary Fig. 13a, b). However, only CXCR4 knockdown blocked *SNAI2* induction by SDF1α with CXCR7 knockdown having little effect (Fig. 7e). Similar results were observed with a second set of siRNA against each receptor (Supplementary Fig. 13a, c). AMD3100, a small molecule inhibitor specific for CXCR4, similarly reduced sprouting (Supplementary Fig. 13d), and this reduction was rescued with Slug overexpression in EC (Fig. 7f and Supplementary Fig. 13d).

Finally, we found that SDF1α treatment increased ERK5 phosphorylation, and this was blocked by both AMD3100, and XMD8-92, an ERK5 inhibitor (Fig. 7g and Supplementary Fig. 14a). Moreover, siRNA knockdown of CXCR4 (but not CXCR7) also prevented ERK5 phosphorylation (Fig. 7h and Supplementary Fig. 14b), and XMD8-92 treatment prevented Slug induction by SDF1α (Fig. 7i). Taken together, these data indicate that the SDF1α-CXCR4 pathway stimulates angiogenesis by phosphorylating ERK5 to induce Slug expression.

## Discussion

Slug's role as a master regulator of EMT in normal and neoplastic epithelial cells has been extensively studied[16,20,56]; however, only recently has a potential role for Slug in EC been revealed through studies in an in vitro sprouting angiogenesis model[19]. Studies in vivo, however, had failed to note a significant vascular phenotype in Slug germline knockout (SlugKO) mice. In this study, we reconcile these findings by performing a focused analysis of vascular growth and remodeling in the SlugKO mice, and report that endothelial Slug controls both developmental and pathological angiogenesis in vivo. Moreover, using a combination of human cell-based 3D in vitro assays and various in vivo mouse models, we show that upon induction by SDF1α-CXCR4 signaling, Slug regulates vascular morphogenesis by modulating Notch-mediated VEGFR2 expression and inducing a partial EndoMT phenotype in angiogenic EC.

Using a syngeneic mouse tumor model, we found that Slug deficiency led to reduced tumor angiogenesis, and in many cases, this was enough to completely halt tumor growth. Previous studies have shown that cancer cell-expressed Slug promotes tumor growth[27,56,57], however Slug's role in EC is less clear. Some

groups[57] (but not others[58]) have observed reduced tumor size in SlugKO animals, but none specifically assessed the effect of Slug deletion on tumor angiogenesis. It is likely that differences in mouse genetic background, the particular model used, and the presence or absence of immune cells have contributed to the variability. To address this, we compared the extent of tumor vasculature in SlugKO animals using three different syngeneic (Slug-expressing) tumor cell lines, each with different origins and tumorigenicity[28–30]. Consistent with earlier studies[57], we also observed reduced tumor growth across all three cell lines in the SlugKO mice, associated with greatly reduced tumor vasculature. EC infiltration and tumor angiogenesis are also drastically reduced in a mock tumor model (i.e. in vivo Matrigel plug assay). Therefore, we conclude that the slowed tumor growth in SlugKO animals is a result of reduced tumor angiogenesis.

We also determined that Slug is important during vessel development. Endothelial Slug deficiency significantly (if transiently) delayed growth of both primary and secondary vascular plexus in neonatal retinas of SlugKO animals. Consistent with our previous in vitro findings, this was partly due to impaired EC migration[19]. We were able to partially rescue this phenotype using selective Slug re-expression in vascular endothelium. This role for endothelial Slug in driving angiogenesis is apparently non-redundant with Snail (unlike in embryonic heart valve formation[22]) since Snail is expressed in retinal EC and its expression was unchanged in SlugKO mice. Further experiments are necessary to determine the relative contributions of Slug vs. Snail in developmental sprouting angiogenesis.

RNA-seq analysis suggests that Slug controls several downstream pathways. Consistent with its function in other cell types[16,56,59], Slug dependent pathways include regulators of cell morphology, junctional and matrix adhesions, proliferation, and TGFβ signaling—all likely components of an EndoMT process. Indeed, EMT was among the top-ranked gene ontology terms associated with this dataset. Genes diagnostic of a mesenchymal transition including *SPARC*, *ACTA2*, *FSCN1*, and *DDR2* are all upregulated during normal sprouting and upon Slug overexpression, while negative regulators of EMT including *FOXP4* and *HHIP* are downregulated by Slug[16,60]. We focused on Slug regulation of cell–cell junction proteins, which was confirmed by both RNA-seq and immunostaining. Junction destabilization is an important feature of EMT, and may contribute to the disconnected vessel fragments/isolated EC observed in the SlugOE bead assay. Conversely, the observed reduction in vessel permeability in SlugKO mice could relate to enhanced cell junction in these animals.

The reduction of *CDH2*, a gene often upregulated during EMT[16], in SlugOE EC might suggest a context-specific difference between EndoMT and EMT, as it is normally expressed in EC and plays numerous roles[61,62]. Thus, a cadherin-switch[63], rather than the simple loss or gain of *CDH2*, might be a better indicator of an EndoMT.

In aggregate, the set of genes regulated by Slug in EC provides strong support for the idea that Slug drives EC toward a partial EndoMT, a process crucial for angiogenic sprouting[13]. Several

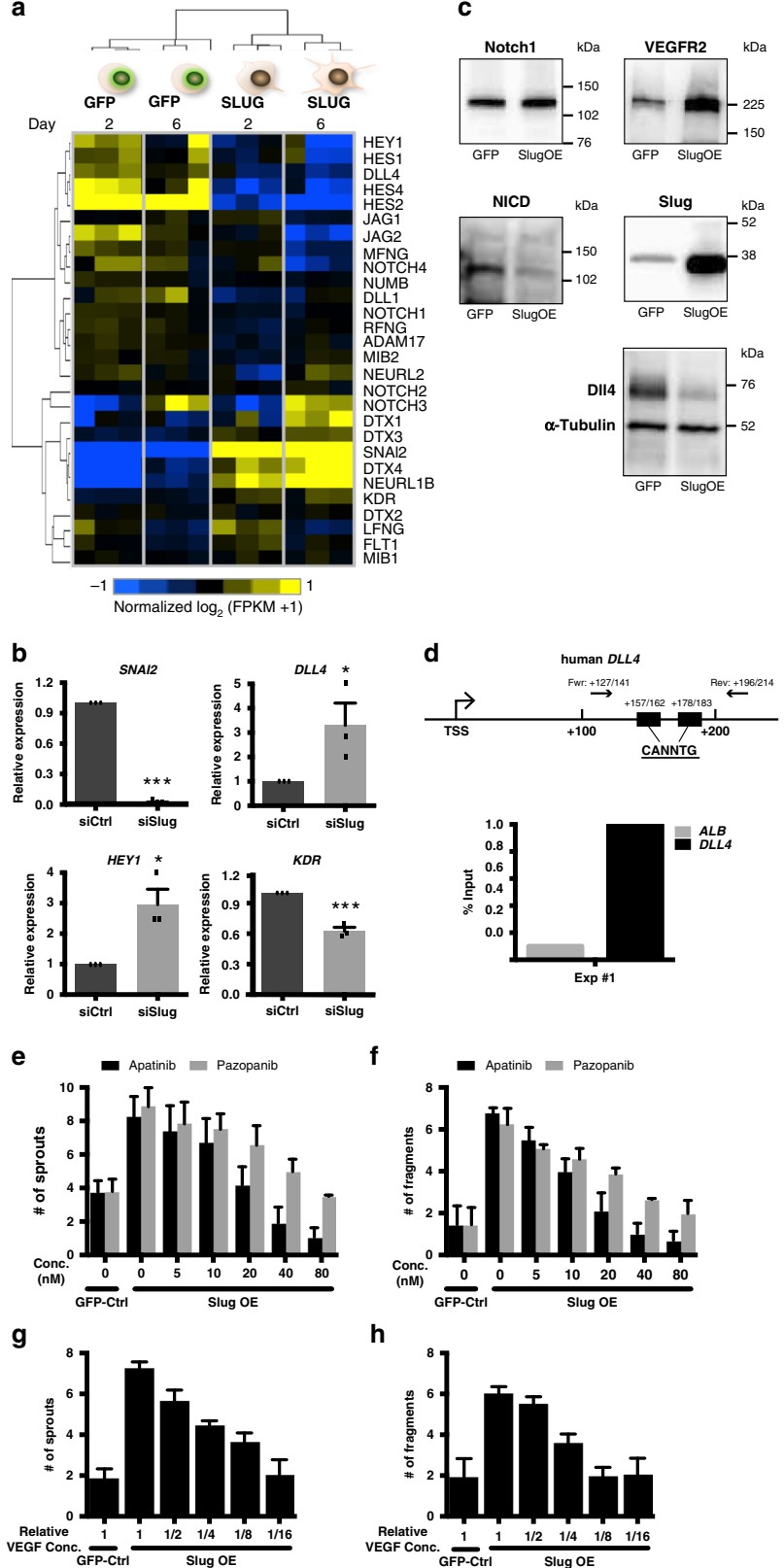

studies have reported on Slug expression in vivo during complete EndoMT[64–66], wherein EC take on a full mesenchymal phenotype and separate from their neighbors and migrate as individual cells. We believe our data best fit the idea that during sprouting angiogenesis, EC undergo a partial EndoMT, wherein a subset of classic EMT markers are induced, leading to transient and reversible adoption of mesenchymal characteristics that enable sprouting. Our data further suggest that this relies upon endothelial Slug, and that Slug levels determine the extent of commitment towards a mesenchymal identity—i.e. whether cells will undergo partial or full EndoMT. Studies are ongoing to test this hypothesis.

**Fig. 5 Slug regulates EC sprouting via Notch-mediated VEGFR2 expression. a** Heat map for differentially expressed Notch pathway components in GFP control and SlugOE EC on day 2 and day 6 of the fibrin-gel bead assay. **b** qPCR analysis of Slug and Notch-related target genes in control ($n = 3$ cell lines) and Slug ($n = 3$ cell lines) siRNA-treated EC. *SNAI2* $p < 0.0001$, *DLL4* $p = 0.0325$, *HEY1* $p = 0.018$, *KDR* $p = 0.0007$. **c** Western blot analysis of Slug and Notch-related target genes in GFP and SlugOE EC. **d** Top: human Dll4 promoter region schematic. Bottom: ChIP qPCR analysis showing enrichment of Slug binding to the *DLL4* promoter region in SlugOE EC (representative of three replicates). **e** Number of sprouts per bead was quantified for untreated GFP-control EC and SlugOE EC treated with either Apatinib or Pazopanib at different concentrations ($n = 30$ beads per condition from three experiments). **f** Number of disconnected fragments around each bead was quantified for untreated GFP-control EC and SlugOE EC treated with either Apatinib or Pazopanib at different concentrations ($n = 30$ beads per condition from three experiments). **g** Number of sprouts per bead was quantified for GFP-control EC and SlugOE EC fed with media containing different levels of VEGF ($n = 30$ beads per condition from three experiments). **h** Number of disconnected fragments around each bead was quantified for GFP-control EC and SlugOE EC fed with media containing different levels of VEGF ($n = 30$ beads per condition from three experiments). Data represent mean ± SEM. Two-tailed unpaired equal-variance *t*-test. *$p < 0.05$, **$p < 0.01$, ***$p < 0.005$. Source data are provided as a Source Data file.

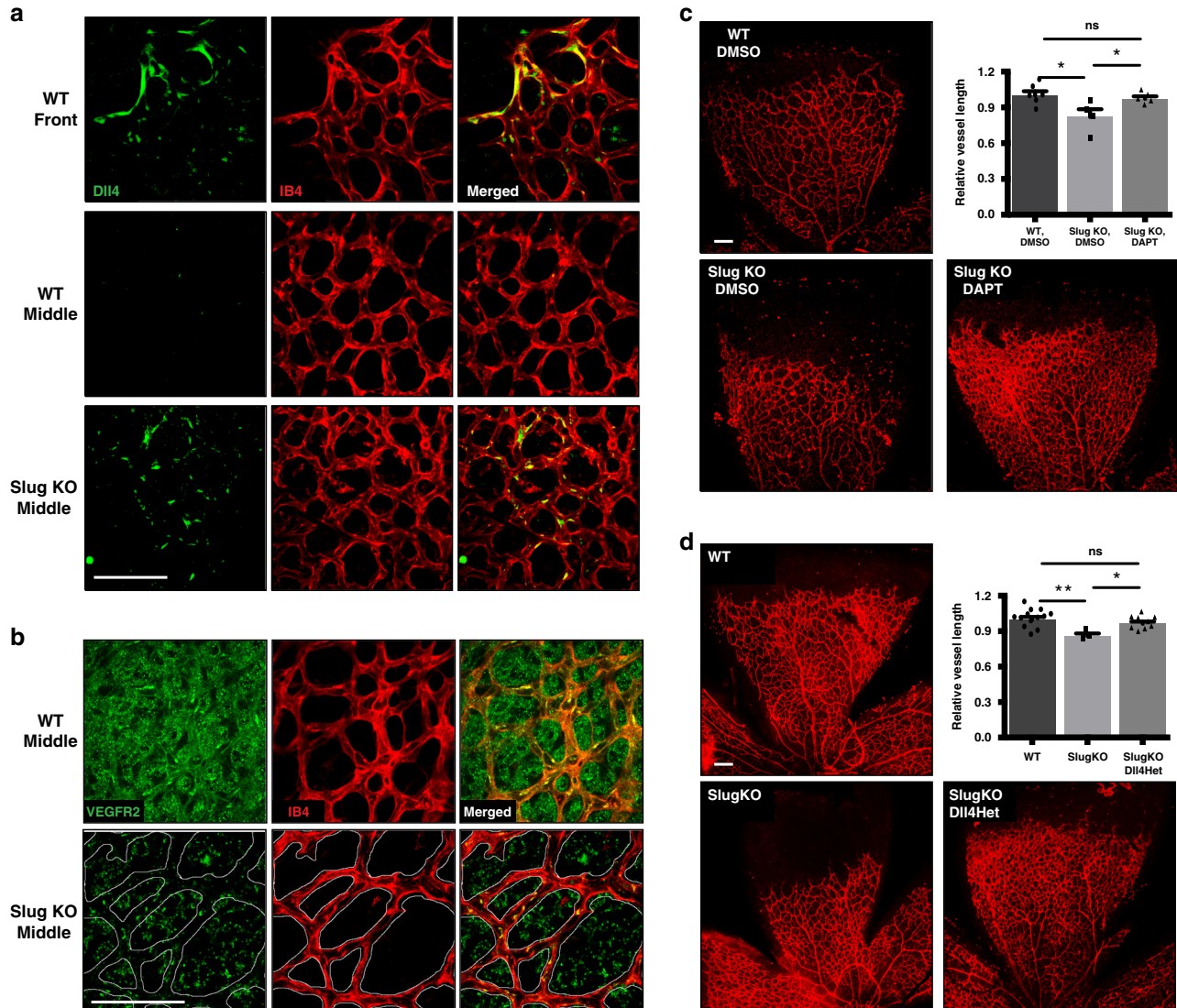

**Fig. 6 Notch inhibition rescues delayed vascular expansion in SlugKO mouse retina. a** Immunofluorescent staining showing localization of Dll4 expression at the angiogenic front and the middle of the network in WT and SlugKO mice, respectively. Scale bar: 100 μm. **b** Immunofluorescent staining showing reduced VEGFR2 expression in SlugKO retinas compared to the WT. Scale bar: 100 μm. **c** IB4 staining of P5 retinas from WT and SlugKO mice treated with DMSO control or DAPT via subcutaneous injection. Scale bar: 100 μm. Quantification shown in top right corner (WT-DMSO $n = 6$ animals, SlugKO-DMSO $n = 4$ animals, $p = 0.03$. SlugKO-DAPT $n = 5$ animals, $p = 0.045$). **d** IB4 staining of P5 retinas from WT, SlugKO and SlugKO/Dll4Het. Quantification shown in top right corner (WT $n = 13$ animals, SlugKO $n = 3$ animals, $p = 0.007$. SlugKO/Dll4Het $n = 9$ animals, $p = 0.014$). Scale bar: 100 μm. Data represent mean ± SEM. Two-tailed unpaired equal-variance *t*-test. *$p < 0.05$, **$p < 0.01$, ***$p < 0.005$. Source data are provided as a Source Data file.

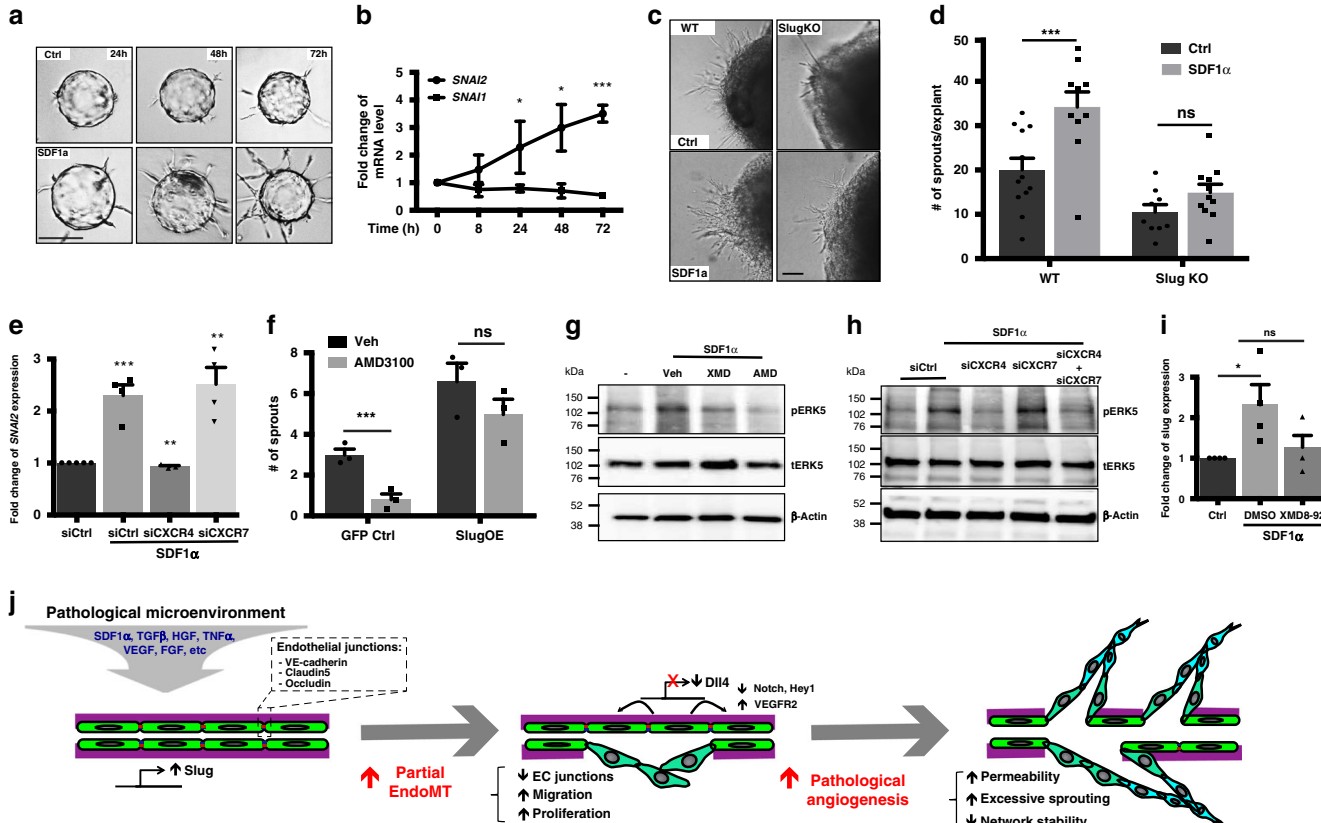

**Fig. 7 Slug is an SDF1α downstream effector during angiogenesis. a** SDF1α treatment leads to early and increased tip cell formation and sprouting in the fibrin-gel bead assay. Scale bar: 100 μm. **b** SDF1α treatment leads to activation of *Snai2* but not *Snai1* transcription ($n \geq 3$ cell lines per condition, see source data for detail. 24 h $p = 0.04$, 48 h $p = 0.012$, and 72 h $p = 0.003$). **c** SDF1α induces increased sprouting in WT ($p = 0.003$) but not SlugKO mouse retina explants. Scale bar: 100 μm. **d** Quantification of the number of sprouts per retina explant (WT-Ctrl $n = 12$ animals, WT-SDF1α $n = 10$ animals, SlugKO-Ctrl $n = 9$ animals, SlugKO-SDF1α $n = 11$ animals). **e** qPCR analysis of Slug induction by SDF1α in EC treated with siCtrl ($n = 4$ cell lines, $p = 0.0002$), siCXCR4 ($n = 3$ cell lines, $p = 0.004$), or siCXCR7 ($n = 4$ cell lines, $p = 0.001$). **f** Slug overexpression in EC rescues AMD3100 inhibition of sprouting. ($n = 3$ cell lines for all conditions, $p = 0.0057$). **g** SDF1α activation of ERK5 phosphorylation and blocking by inhibition of ERK5 (XMD8-92) or CXCR4 (AMD3100). **h** ERK5 phosphorylation downstream of SDF1α is abolished by CXCR4 knockdown but not CXCR7 knockdown. **i** SDF1α induction of Slug ($p = 0.0037$) is blocked by the ERK5 inhibitor XMD8-92 ($n = 4$ cell lines for all conditions). **j** Model for Slug-mediated pathological angiogenesis. Data represent mean ± SEM. Two-tailed unpaired equal-variance *t*-test. *$p < 0.05$, **$p < 0.01$, ***$p < 0.005$. Source data are provided as a Source Data file.

We also observed strong downregulation of Notch pathway components in SlugOE EC, and proved direct binding of Slug to the E-box sequences in the promoters of the genes coding for Notch ligands Dll4 and Jag2, consistent with recent studies in mouse[44] and human[45]. Moreover, consistent with our previous report[46], reduced *DLL4* lowers Notch signaling and the expression of the Notch target *HEY1*[43,46–48], resulting in increased KDR expression in SlugOE EC. As expected, VEGF signaling inhibition both corrected the hypersprouting phenotype and reduced the number of disconnected fragments (an indicator of aggressive EndoMT) in SlugOE EC. Finally, reduced Notch signaling, either by DAPT treatment or Dll4 haploinsufficiency, in the neonatal mouse retina rescued the vascular defects caused by Slug deficiency, confirming the close relationship between Slug and the Notch pathway during angiogenesis. While previous studies have shown Slug to be a downstream effector of Notch signaling during EMT[67,68], to our knowledge, this is the first study to show Slug regulation of Notch through Dll4 suppression.

In epithelial cells, Slug expression is regulated by several factors including TGFβ, Wnt, HGF, and TNFα[16]. Little is known about the upstream regulation of Slug in EC. Here, we found that SDF1α, a key angiogenic factor[51,52] expressed by fibroblasts[33], operates mainly through its canonical receptor CXCR4 to upregulate Slug. Although SDF1α's auxiliary receptor CXCR7 was recently shown to promote tumor angiogenesis[53,54], and its loss blocks sprouting angiogenesis in the fibrin-gel bead assay, CXCR7's effects on angiogenesis are independent of Slug. Instead, SDF1α-CXCR4 activation in EC drives *SNAI2* transcription through downstream ERK5[69].

In aggregate, our data identify EC-expressed Slug as a critical regulator of angiogenesis, particularly in the pathological setting. Upon activation by SDF1α (and likely other factors), Slug downregulates Dll4, which reduces Notch activity and increases VEGFR2 expression in EC. Slug expression also initiates a partial EndoMT program, which we propose pushes EC into an activated, mesenchymal-like state, characterized by reduced junctional adhesion as well as enhanced proliferation and migration (Fig. 7j). Thus, Slug is a key regulator of pathologic angiogenesis and may prove to be a useful anti-angiogenesis/anti-tumor target.

## Materials and methods

**Mice.** All animal studies were approved by and conducted in compliance with University of California, Irvine IACUC regulations. All animals are housed at 22 °C ambient temperature, 40–60% humidity and are on a 12-h dark, 12-h light cycle. The B6;129S1-Snai2[tm2Grid/J] (Slug knockout, SlugKO) mice[22] were originally obtained from The Jackson Laboratory. These mice were then backcrossed with C57BL/6J mice (Jackson Laboratory) once to establish the colony. All the SlugKO mice used in this study were derived from this colony unless otherwise noted. The B6.FVB-Tg(Cdh5-cre)[7Mlia/J] mouse (VEcad-Cre) was a generous gift from Dr.

Luisa Iruela-Arispe from Northwestern University. The SlugKO, VEcad-Cre mice used for endothelial-specific rescue experiment were generated by first crossing the SlugKO mice with VEcad-Cre mice to generate the double heterozygous (Slug$^{+/-}$, VEcad-Cre$^+$), and then intercrossed to obtain SlugKO, VEcad-Cre$^+$ males. These SlugKO, VEcad-Cre$^+$ males were then crossed with the Slug$^{+/-}$, VEcad-Cre$^+$ females to generate all the SlugKO, VEcad-Cre$^+$ and SlugKO, VEcad-Cre$^-$ mice for the experiment. Dll4$^{tm1Jrt/ICR}$ (Dll4 heterozygous, Dll4Het, Dll4 $^{+/-}$) mice[70] were obtained from the Canadian Mouse Mutant Repository (CMMR) and were maintained on an ICR background. The SlugKO mice were crossed with the ICR mice once and then crossed with Dll4Het mice to generate the double heterozygous (Slug$^{+/-}$, Dll4$^{+/-}$). These mice were then intercrossed to obtain the SlugKO, Dll4$^{+/-}$ mice used to examine the rescue effect from the loss of Dll4 in SlugKO mice. Wild-type (WT) littermates of the respective backgrounds were used as control in each experiment unless otherwise noted.

**Subcutaneous tumor model**. Mouse syngeneic colorectal cancer cell lines MC-38 (NCI), CMT-93 (ATCC), and B16 (ATCC) were expanded in vitro prior to implantation. Cancer cells were mixed with Matrigel (BD #354234) and injected subcutaneously on the right flank of 20–24 week old male and female WT and SlugKO mice at 5 million (MC-38), 10 million (CMT-93), and 2 million (B16) cells per implant. All tumors were allowed up to 6 weeks to develop during which tumor size were measured daily or until they reached the maximum size allowed by the IACUC protocol. The tumors were then harvested and fixed in 10% formalin overnight at room temperature before paraffin embedding and sectioning for IHC analysis.

**Matrigel plug in vivo angiogenesis assay**. Growth factor reduced Matrigel (BD #356231) was mixed with 20 U/mL Heparin, 400 ng/mL recombinant mouse bFGF (Shenandoah), and 400 ng/mL recombinant mouse VEGF 165 (Shenandoah). WT and SlugKO male and female mice at 20–24 weeks of age were used, consistent with the mice used for the subcutaneous tumor model. A 400-µL bolus of Matrigel mix was injected subcutaneously on the ventral lower abdominal area of the mice, close to the mid-line. For some experiments, mice received unilateral injection of Matrigel containing isolated wild-type mouse fibroblasts[34] (200,000 cells per plug) and acellular Matrigel injected into the contralateral control side. In all experiments, Matrigel plugs were harvested 7 days post-injection, and fixed in 10% formalin overnight at room temperature. The plugs were then paraffin embedded and sectioned for IHC analysis. Each experiment was repeated three times with at least 5 mice per genotype each time.

**Mouse neonatal retina vascular development model**. Postnatal mice were euthanized according to IACUC-approved protocols for P5-15 mice. In all studies, following euthanasia of experimental animals, mouse eyes were enucleated and immediately fixed in 4% PFA on ice for up to 4 h. Eyes were then washed thoroughly with cold PBS and dissected. Obtained retinas were then permeabilized with PBS containing 0.3% Triton-X 100 and blocked with 5% BSA overnight at 4 °C. Retinas were incubated with primary antibodies for 24–96 h at 4 °C and washed before secondary antibodies were incubated for up to 48 h at 4 °C or 1 h at room temperature. Retinas were then washed thoroughly, post fixed with 4% PFA, and mounted in Prolong Gold (Thermo Fisher P36930). For EdU studies, P6 pups received intraperitoneal injection of 100 mg/kg EdU in sterile PBS, and euthanized 2 h later for retinal dissection and analysis using Click-iT EdU Imaging Kit (Invitrogen, C10337). For the DAPT studies, mice received subcutaneous injection of DAPT (Selleckchem, 100 mg/kg per mouse, 20 ul per injection, dissolved in sterile corn oil) or vehicle on P3 and P4, once daily, prior to euthanasia on P5. All the experiments were repeated with at least 3 litters of animals, with at least two mice per genotype per litter.

**FACS sorting of retinal EC**. Neonatal mice aged between P8 and P12 were used in this experiment. Mice were euthanized and unfixed retinas from 2 to 4 pups of WT, Slug heterozygous or SlugKO mice were dissected into cold PBS and pooled. After dissection, retinas were placed into 1 mL of digestion buffer containing 1 mg/mL collagenase type II (Worthington) in 20% FBS low-glucose DMEM and incubated at 37 °C for 20 min. After digestion, the cells were passed through a 70-µm cell strainer, centrifuged at 150 × g for 5 min at room temperature and resuspended in staining buffer containing 1% BSA. Cells were stained with primary CD31 antibody for 30 min on ice, washed, and stained with fluorescently conjugated secondary antibody for 15 min on ice in the dark. After thorough washing, CD31$^+$ cells were sorted using BD Aria II.

**EC specific re-expression of Slug in SlugKO mouse retina**. The Switch lentiviral construct was generated from the Cre Reporter backbone from Niels Geijsen (Addgene plasmid #62732) and lentiviral vector pCDH (CD521A-1; System Biosciences). The full-length mouse Slug cDNA was cut from pTK-Slug (Addgene plasmid #36986) and cloned into the pCDH vector. The mSlug-T2A-GFP expression cassette was then amplified from this construct by PCR for subsequent cloning. Next, the GFP-IRES-Puromycin Resistance expression cassette from the Cre Reporter plasmid was cut out using HpaI and ClaI and replaced with either GFP (Switch-GFP) or mSlug-T2A-GFP (Switch-mSlug-GFP) using Cold

Fusion Cloning Kit (SBI, MC010B-1). Lentivirus from both Switch-GFP and Switch-mSlug-GFP were generated in 293T cells and concentrated up to 1000-fold by ultra-centrifugation at 100,000 × g for 90 min. Functional viral titer was tested using FACS sorting method and the viral titers used were at least $1.5 × 10^8$ FU/mL.

SlugKO,VEcad-Cre$^+$ or SlugKO,VEcad-Cre$^-$ P6 mice received Switch mSlug-GFP virus via a single retro-orbital injection (15 µL of virus + 5 µL 0.1% Evans Blue to mark injection); some littermate controls received control (Switch GFP) virus. After 7 days, mice were euthanized, their retinas were harvested, and stained using IsolectinB4-Alexa647 (Vector Labs). The secondary plexus of their retinal vasculature was imaged and vascularization index (percentage of retinal area occupied by vessels) was quantified using the grid-counting method detailed in "Microscopy and Image Processing" below. SlugKO littermates without injection were also included to control for the injection procedure and no obvious damage to the retinal vasculature was observed.

**Miles Assay for vascular permeability**. Age-matched young WT and SlugKO male and female mice (6–12 weeks) were shaved 24 h prior to the experiment. On the day of experiment, mice were first anesthetized by intraperitoneal injection of ketamine/xylazine, and then received Evans Blue dye (1% w/v in 0.9% PBS, 100 µL) via retro-orbital injection. After 15–30 min, each mouse received intradermal injections of 20 µL mVEGF (2.5 ng/µL) unilaterally on one flank, or equal volume of PBS on the contralateral side, using a 31 gauge needle. Mice were euthanized 30 min post-injection and skin at each injection site, roughly 1 cm$^2$ in size were excised and placed into an open 1.5 mL tube. The skin samples were dried and the Evans Blue dye was extracted into 250 µL formamide. Following centrifugation at 30,000 × g for 5 min at room temperature, 100 µL of formamide containing the Evans Blue dye was added to a 96-well plate and the absorbance was read at 620 nm with a reference reading at 740 nm. The relative absorbance was calculated and plotted.

**Mouse retina explant model**. P4 pups were euthanized and sterilized with ethanol. Eyes were immediately enucleated and the retinas were dissected out in cold endothelial basal medium (EBM2, Lonza) under sterile condition. The retina pieces were serum starved overnight at 37 °C and then embedded into extracellular matrix mixture (70% Collagen I + 30% Matrigel). Retina tissues were cultured for 6 days with 3% EBM2 (with or without SDF1α), where media was replaced every other day. The experiment was repeated three times with retina pieces from at least two mice per genotype each time.

**Cell culture and siRNA transfection**. HUVEC were isolated in house and maintained in M199 (Gibco) supplemented with 10% FBS. Prior to experimentation, HUVEC were switched to culture in complete EGM2 Bullet Kit medium (Lonza CC-3156 & CC-4176). For gene knockdown studies, siRNA was transfection into HUVEC with Lipofectamine 2000 (Invitrogen 11668019) for 4 h and allowed to recover overnight in fresh EGM2 medium for subsequent study using the fibrin-gel bead assay. The next day, cells were coated onto cytodex beads and embedded in fibrin gel with or without fibroblasts. All assays were terminated no later than 6 days post-transfection. Knockdown efficiencies were evaluated 4 days post-transfection using qPCR.

Individual siRNAs and concentrations used are listed here: Stealth RNAi siSlug (50 nM, ThermoFisher, HHS109995), Silencer Select siSlug (50 nM, ThermoFisher, s13127), Stealth RNAi Negative Control (50 nM, ThermoFisher 12935114), siCXCR4 (25 nM, ThermoFisher S532662, S532663), siCXCR7 (12.5 nM, ThermoFisher S94, S95), Silencer Select Negative Control 1 (25 nM, ThermoFisher 4390843), and Silencer Select Negative Control 2 (12.5 nM, ThermoFisher 4390847).

**3D Fibrin bead angiogenesis assay**. The 3D in vitro bead assay was performed and analyzed as previously described[19]. Briefly, HUVEC were coated onto Cytodex 3 beads (GE 17048501) and embedded in a fibrin gel (MP Biomedicals) in 24-well plates at ~150 beads per well. Normal human lung fibroblasts (NHLF) were then seeded on top of the gel at 20,000 cells per well. The cultures were maintained in EGM2 for the number of days specified. Inhibitor treatments were started 2 days after embedding. For SDF1α induction of Slug expression, EC-coated cytodex beads were embedded without the fibroblasts and the medium was changed daily. Each experiment was repeated with at least three different HUVEC lines. At least 10 beads were analyzed for each condition in each experiment.

**Vascularized micro-organs model**. VMO were established in a microfluidic platform as previously described[38]. Briefly, transduced HUVEC (control-GFP or SlugOE-GFP) and NHLF were harvested and resuspended at a 1:1 ratio, for a final density of $1 × 10^7$ cells/mL in 10 mg/mL fibrinogen solution with thrombin. The cell-matrix suspension was then quickly loaded into the tissue chambers of the microfluidic platform. After gel is polymerized, cell culture medium EGM2 (Lonza) is added to the inlet and outlet medium reservoirs to establish hydrostatic pressure and to generate laminar flow along the microfluidic channels and interstitial flow across the tissue chambers. Cell culture medium was replaced every other day. After 5 days of culture, control-GFP and SlugOE-GFP VMOs were subjected to

vascular perfusion tests with 70 kDa Rhodamine B dextran (25 μg/mL, Thermo Fisher) to ensure patency. Each experiment was repeated with at least three different HUVEC lines.

**RNA-sequencing and data analysis**. The 3D fibrin-gel bead assay was performed using either control GFP or Slug overexpressing GFP (SlugOE) HUVEC with three biological replicates each. The EC were harvested at days 2 and 6 post-embedding by serial trypsinization using 5% trypsin. First, the NHLF monolayer growing on top of the fibrin gel along with the top layer of the gel were lifted with a 3-min incubation with 5% trypsin under a heat lamp while agitated. Then, the remaining gel containing the EC-coated beads were thoroughly washed with Hanks buffer three times, collected into a 15-mL Falcon tube with fresh 5% trypsin, and incubated at 37 °C for up to 5 min to release the EC-coated beads. The EC-coated Cytodex beads were then separated from the supernatant by brief centrifugation and further treated with collagenase IV (Worthington) to release EC from the beads, and then filtered through a 100-μm filter top tube to remove the beads. After another 5-min centrifugation at $150 \times g$, EC were resuspended in 300 μL RLT buffer (Qiagen 74104) and incubated at 55 °C with Protease K (Qiagen 19131) for 15 min. The RNA was then isolated following the manufacturer's protocol using a Qiagen RNA Mini Kit (Qiagen 74104).

RNA integrity (RIN) was measured for all samples using a Bioanalzyer Agilent 2100. All sequencing libraries analyzed were generated from RNA samples with a RIN score ≥9. The Illumina TruSeq mRNA stranded protocol was used to obtain poly-A mRNA from all samples. A total of 200 ng of isolated mRNA was used to construct RNA-seq libraries, and these were quantified and normalized using the Library Quantification Kit from Kapa Biosystems and sequenced as single reads on an Illumina HiSeq 2500 platform. At least 30 million reads per sample were sequenced.

Quality control was performed on data using FASTQC (v. 0.11.2) and reads were trimmed using Trimmomatic (v.0.32) with Illumina TruSeq adapter sequences using a PHRED quality score 15 and minimum length 20 bases. Trimmed reads were then aligned to the human hg19 reference genome using the Ensembl GRCh37 annotations and post processed using Tophat2 (v.2.0.12), Bowtie2 (v.2.2.3), and Samtools (v.0.1.19). Expression levels were quantified both with FPKM (Fragment Per Kilobase per Million mapped reads) using Cufflinks (v. 2.1.1) and with raw counts using HTSeq (v.0.6.1p1.). Differential analysis was done using DESeq2 and significant genes were considered ($p$-value < 0.05, 2-fold change). K-means clustering was performed to extract dynamic gene clusters based on differential expression between GFP and Slug knockdown time-points. Gene Set Enrichment Analysis (GSEA) was performed to identify significantly enriched gene sets across sample comparisons. Additional gene ontology analysis was performed for each cluster and reported using Metascape. Clustering was performed in R and heatmaps were generated using JavaTree.

**Chromatin immunoprecipitation**. HUVEC were grown to confluency and cross-linked for 10 min at room temperature by directly adding formaldehyde in cold PBS into the culture dish for a final concentration of 1%. Crosslinking was stopped by adding glycine to a final concentration of 0.125 M and cells were collected on ice, pelleted, and snap froze in liquid nitrogen. Before sonication, pellets were thawed on ice, lysed in 5 mM PIPES buffer with 85 mM KCl, 0.5% NP-40, and Protease Inhibitor Cocktail (Roche) and passed through a 20-gauge needle repeatedly to break open cells. After centrifuging at $150 \times g$ for 10 min at 4 °C, supernatant was removed and RIPA buffer was added to lyse the nuclei. Chromatins were then sonicated on ice to obtain chromatin fragments that are ~300 bp. To attach antibody to the magnetic beads, the sheep anti-rabbit beads (Invitrogen 11204D) were first blocked with 5 mg/mL BSA, then incubated with polyclonal rabbit anti-Slug antibody (Santa Cruz Biotch) on a rotator overnight at 4 °C. 100 μL chromatin was saved as input control and the rest was used to perform ChIP by adding the chromatin to the antibody-bound beads and incubating on a rotator overnight at 4 °C. To reverse crosslinking, beads were washed thoroughly and incubated at 65 °C for 5 h with 1 min of vortexing every 15 min. After centrifuging at $150 \times g$ for 5 min at 4 °C, supernatant was transferred to a new tube and incubated at 65 °C overnight to complete reverse corss-linking. Both input control and immunoprecipitated chromatins were purified with Qiagen PCR purification kit (Qiagen 28104) and concentration was determined using a Qubit Fluorometer.

**Immunohistochemistry, immunocytochemistry, and western blot**. For Immunohistochemistry, tissues were fixed in 10% formalin, embedded in paraffin, and sectioned. Sections were deparaffinized, rehydrated, and treated with sodium citrate unmasking solution (Vector Labs) for 20 min in a rice cooker. Sections were blocked with hydrogen peroxide for 30 min, and then with 5% rabbit serum and 2% BSA in PBS for 1 h. Primary antibodies: rat anti-mouse CD31 (Dianova, DIA-310) or mouse anti-Ki67 (Roche/Venatana 790–4286) was added and incubated overnight at 4 °C, followed by washing in 0.1% TBS-T for 20 min. Secondary antibodies were added for 1 h, followed by washing in 0.1% TBS-T, development using ABC Kit (Vector Labs AK5004) and DAB Substrate Kit (Vector Labs SK4100). Cells were counterstained with hematoxylin, dehydrated, and mounted in Cytoseal (Richard-Allen Scientific).

For immunocytochemistry, HUVEC were grown to confluency for 7 days on gelatin-coated Permanox slides (Thermo Fisher 177437). Cells were fixed and permeablized in cold methanol for 20 min and blocked with 5% BSA in PBS. Primary antibodies were added and incubated overnight at 4 °C. After cells were thoroughly washed, secondary antibodies and DAPI were added and incubated for 1 h at room temperature. Cells were washed and mounted in ProlongGold (Thermo Fisher).

For western blotting, cells were collected and lysed directly in 2x SDS buffer. Protein concentration was assessed with protein BCA assay (Thermo Fisher 23235). In all, 20 μg of protein were run on 4–12% gradient gel (Bio-rad 4561094) and transferred overnight at 4 °C. Membranes were blocked with 5% BSA and incubated with primary antibodies overnight at 4 °C. After thorough washing with TBS buffer containing 0.1% Tween20, secondary antibodies were added and incubated at room temperature for 1 h. Blots were developed with Amersham ECL Prime western blot detection reagent (GE RPN2232) and visualized using a Bio-Rad Gel Doc.

**Antibodies, recombinant proteins, and inhibitors**. These were from the sources indicated and were used as described in the text and legends.

Antibodies for immunofluorescence: fluorescein/Rhodamine/Alexa647 labeled GSL I isolectin B4 (1:500, Vector Labs), rabbit anti-Slug (1:100, Cell Signaling 9585), mouse anti-NG2 (1:100, Millipore 5384), mouse anti-GFAP (1:100, Millipore 360), mouse anti-rabbit anti-Claudin5 (1:400, Abcam 53765), rabbit anti-VE-cadherin (1:400, Enzo ALX-210-232), mouse anti-CD31 (1:50, Dako 0823), goat anti-Dll4 (1:50, R&D AF1389), rabbit anti-VEGFR2 (1:100, Cell Signaling 2479), goat anti-Alexa488 (1:500, Invitrogen A11034), goat anti-rabbit-A594 (1:500, Invitrogen R37117), goat anti-mouse-A647 (1:500, Invitrogen A21235), and donkey anti-goat-A647 (1:500, Invitrogen A21447). Antibodies for western blot: mouse anti-alpha-SMA (1:1000, Dako M0851), rabbit anti-SPARC (1:1000, Cell Signaling 8725), mouse anti-CCND1 (1:1000, Santa Cruz Biotech, sc-8396), rabbit anti-p21 (1:1000, Cell Signaling 2947), rabbit anti-NICD (1:1000, Cell Signaling 4147), rabbit anti-Notch1 XP (1:1000, Cell Signaling 3608), rabbit anti-Dll4 (1:1000, Cell Signaling 2589), rabbit anti-Slug (1:500, Cell Signaling, 9585), rabbit anti-VEGFR2 (1:1000, Cell Signaling 2479), rabbit anti-Hey1 (1:500, Abcam 22614), rabbit anti-pERK5 (1:1000, Cell Signaling 3371), rabbit anti-ERK5 (1:1000, Cell Signaling 3552), rabbit anti-alpha-Tubulin (1:2000, Cell Signaling 2144), rabbit anti-GAPDH (1:2000, Cell Signaling 5174), rabbit anti-beta-Actin-HRP (1:2000, Cell Signaling 5125), anti-rabbit-HRP (1:2000, Cell Signaling 7074), and goat anti-mouse-HRP (1:2000, Santa Cruz Biotech sc-2031). Antibody for FACS sorting: rat anti-mouse-CD31 (1:200, Dianova, DIA-310) and goat anti-rat-A488 (1:500, Invitrogen, A-11006). Antibody for immunohistochemistry: rat anti-mouse-CD31 (1:75, Dianova, DIA-310) and mouse anti-Ki67 (Roche/Venatana 790–4286). Antibody for ChIP: rabbit anti-Slug (1:50, Santa Cruz Biotech 15391).

Recombinant proteins: hSDF1α (Shenandoah 100-20), mVEGF-165 (Shenandoah 200-34), and mFGFb (Shenandoah 200-12).

Inhibitors: DAPT (GSI-IX) (Selleckchem S2215), Apatinib (Selleckchem S3012), Pazopanib (Selleckchem S2221), AMD3100 (Selleckchem S8030), and XMD8-92 (Tocris 4132).

**Microscopy and image processing**. Bright-field images were taken with an Olympus IX70 inverted microscope. Low magnification confocal images were taken with a Nikon Eclipse Ti inverted confocal microscope and high-magnification confocal images and z-stacks were taken with a Leica SP8. Images were analyzed with ImageJ software and processed in Photoshop CC2015 (Adobe).

For retinal vessel length, the average of three measurements was taken from the center of the vasculature to the leading edge for each leaflet. To calculate retinal vascularization index, a randomly generated 80 μm² grid was overlaid onto the image, and the number of crosses intersecting with blood vasculature was counted as a ratio of all intersections to give the vascularization index. For leading edge vascularization index, a region of interest 100 μm × 300 μm was applied at the vascular front. For expression of Slug in various cell types in the retina, 30 μm z-stacks were taken at different regions of the primary plexus and co-localization of Slug with nuclear staining was determined via orthogonal projection and slice analysis (Leica, LAS X). Retinal pericyte coverage was similarly quantified by comparing the number of crosses intersecting NG2$^+$ pericytes with those intersecting vessels (IB4$^+$). For quantification of vessel proliferation, vessel plexi were segregated into radial arterial or venous segments centering around either artery (marked by an αSMA$^+$ large vessel) or vein (αSMA$^-$ large vessel) and with boundaries equidistant to the next adjacent arterial or venous vessel. Within each radial segment, the number of EdU$^+$/CD31$^+$ double-positive nuclei were counted and normalized to overall vessel area.

For cell proliferation in EC cultures, number of EdU$^+$ cells were calculated as a percentage of total nuclei present and averaged across several high-magnification images.

For in vivo Matrigel assays, multiple low-magnification images were taken to include the entire Matrigel section. Percentage of EC area was determined by dividing the total CD31$^+$ area by total Matrigel area for each plug. The percentage vessel area was determined by measuring the cross-sectional area contained within CD31$^+$ staining and divided by the total Matrigel area for each plug.

For the syngeneic tumor model, 15 representative regions of interest in the tumor were quantified for percentage of Ki67-positive nuclei using ImageJ. Percentage EC area at the center region of the tumor was determined, where the center region was defined within a 1500-μm radius from the center of the tumor. The necrotic core size was determined by measuring the total acellular area at the center of the tumor.

**Statistical analysis and reproducibility**. For all quantifications, researchers were blinded to experimental conditions. Data were analyzed and graphed using Graphpad Prism 6. All data are presented as mean ± SEM. Unpaired two-tailed Student's *t* tests assuming equal variance were used to determine statistical significance between two groups unless otherwise noted in figure legends. A *p*-value < 0.05 was considered statistically significant. All experiments and treatments, including immunofluorescence staining and western blot, were repeated with three or more experiments or biological replicates (cell lines, devices, and animals) unless otherwise noted.

**Q-PCR primer sequences for human and mouse genes**. See Supplementary Table 1 (human) and Supplementary Table 2 (mouse).

**Reporting summary**. Further information on research design is available in the Nature Research Reporting Summary linked to this article.

## Data availability

All data generated or analyzed during this study are included in this published article (and its supplementary information files). The results of differential-gene-expression analysis results from the RNA-seq experiment are included in the Supplementary Data 1. Raw data are deposited into the GEO database under the accession code GSE154546. Source data are provided with this paper.

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

## Acknowledgements

We thank the entire Hughes lab for suggestions and scientific discussion. We thank Dr. Melanie Oakes, Dr. Jenny Wu, and the UCI Genomic High Throughput Facility for consulting, conducting the RNA-sequencing reactions and initial pipeline analysis. We thank the UCI Transgenic Mouse Facility, and especially Dr. Grant MacGregor, for their help in the design and initiation of the animal experiments, and Vanessa Scarfone at the UCI Bill and Susan Gross Stem Cell Research Center, Flow Cytometry Core for assistance with sorting. We thank the following undergraduate researchers for their help with mouse genotyping and data analysis: Arvind Suresh, Nadia Zarate, Khoa Le, Anton Arce, and Rhogerika Juan. Finally, we thank Dr. Xing Dai, Dr. Matthew Inlay, Dr. Naomi Morrissette and Dr. Marian Waterman at UCI, Dr. Dritan Agalliu at Columbia University, as well as Dr. Rosemary Akhurst at UCSF—all provided invaluable insights and suggestions. This work was supported by US National Institutes of Health [RO1HL60067; to C.C.W.H.]. N.W.H. was supported by an American Heart Association Predoctoral Fellowship [16PRE27530026], a Ruth L. Kirschstein National Research Service Award Individual Predoctoral Fellowship from the US National Institutes of Health [F31HL131381], and by the ARCS Foundation. M.E.Z. was supported by the National Institutes of Health/National Cancer Institute Institutional Training Grant Fellowship, T32CA009054. K.M.W.-R. was supported by a predoctoral award from the American Heart Association. D.S.W and A.E.P were supported by US National Institutes of Health [R01 EY027442, R01 EY013408, core grant P30 EY000331; to D.S.W.]. A.M. and R.N.R were supported by US National Institutes of Health [UM1HG009443; to A.M.]. C.C.W.H. also receives support from the Chao Family Comprehensive Cancer Center (CFCCC) through an NCI Center Grant [P30A062203]. The UCI Experimental Tissue Resource, the Genomics High Throughput Facility, and the Transgenic Mouse Core are also supported by the CFCCC through this award.

## Author contributions

N.W.H. performed the majority of the experiments and wrote most of the manuscript. J.S.F. performed the EdU and αSMA staining and analysis in mouse retinas, Ki67 imaging and analysis in subcutaneous tumor model, and comparison of Slug expression across colorectal cancer cells lines, and wrote the corresponding method sections. M.M.S.H. and L.S.K. performed the immunohistochemistry staining and analysis and wrote the corresponding methods section. D.T.T.P. performed the 3D vascularized micro-organ experiment and wrote the corresponding methods section. R.N.R. performed the RNA-seq differential expression analysis and wrote the corresponding methods section. M.E.Z. performed preliminary experiments with SDF1α in the 3D fibrin-gel bead assay and conducted the initial western blot experiment to examine ERK phosphorylation. K.M. W.-R. performed and analyzed the SlugOE rescuing SDF1α inhibition experiment. A.E.P. performed imaging and analysis of the in vivo EC-specific rescue experiment. N.N.S. performed some of the mouse neonatal retina vasculature analysis. A.M. provided guidance in RNA-seq analysis. D.S.W. and M.A. provided guidance and funding support for A.E.P. and R.N.R. respectively. N.W.H., J.S.F. and C.C.W.H. wrote and edited the manuscript. C.C.W.H. directed the research and provided funding support for the project.

## Competing interests

All authors except C.C.W.H. and D.T.T.P. declare no competing interests. Principal Investigator C.C.W.H. has an equity interest in Aracari Biosciences, Inc, which is commercializing the microfluidic device used in this paper. D.T.T.P. serves as a scientist at Aracari Biosciences and receives stock and financial compensation from Aracari Biosciences. The terms of this arrangement have been reviewed and approved by the University of California, Irvine in accordance with its conflict of interest policies.
