## [Peer Review File · Nature Communications]

Reviewers' Comments:

Reviewer #1:

Remarks to the Author:

In the present ms. the authors analyze the involvement of Slug in angiogenesis, following their previous *in vitro* studies supporting a positive role of Slug (and Snail) in promoting angiogenesis in human endothelial cells (HUVEC) (Welch-Reardon et al., *J Cell Sci*, 2014). They now report the effect of Slug deletion in *in vivo* angiogenesis during retina development and tumor growth using a constitutive Slug KO model. Then, they moved to the *in vitro* system of HUVEC where they manipulate Slug expression (gain and loss of function) and analyze the effect at molecular and cellular level. The authors come to the conclusion that Slug controls angiogenesis by regulating the Dll4-Notch-VEGFR axis, as stated in the title. This study provides some information of interest that in fact complements other studies reporting a similar role, and mechanism of action, for Snail in angiogenesis (Wu et al., *Nat Comm*, 2014). Unfortunately, that previous finding decreases the general interest of the present study for a broad audience. In addition, there are several important concerns with the *in vivo* and *in vitro* data that require further clarification and additional experimentation. In particular, the proposed implication of Slug in mediating a partial endothelial-mesenchymal transition (EndoMT) is not convincingly demonstrated in the *in vivo* Slug KO model. Other concern refers to what extent all *in vitro* data, including the proposed mechanism for Slug in angiogenesis obtained from human EC, can be extrapolated to the *in vivo* mouse model. The ms. also suffer from interpretation of data (in particular gene expression studies) and contains some formal defects. Overall, in my opinion, the ms. does not provide convincing evidence to support the main authors's hypothesis and only provides incremental knowledge on the implication of EMT factors of the Snail family in angiogenesis.

Main points:

1. Tumor data in Slug KO model. The authors analyze the tumorigenic capacity of two syngeneic colorectal cell lines in wt and Slug KO mice. Those cell lines should be described and provide references. Is the proliferation or apoptosis of tumors affected in the absence of Slug? This is an important point requiring clarification in view of the prosurvival action of Slug in other tumor contexts. These studies should be complemented with additional tumor models (like syngeneic breast carcinoma and using orthotopic injections) to prove the generality of the observations.
2. Moreover, the authors should comment on other studies on *in vivo* tumorigenesis in the same Slug KO mice model, as in skin tumorigenesis induced by UV-radiation (Newkirk et al., *Am J Pathol* 171: 1629-39, 2007) or chemical carcinogenesis (Villarejo et al., *Carcinogenesis* 36: 585-97, 2015), with different outcomes. The later study shows that in fact Slug deletion caused increased tumor progression by inducing a pro-inflammatory response. The authors should therefore discuss those data in the context of the present study.
3. Slug expression in *in vivo* samples (tumors, retina) needs improvement. IF data shown in Fig. 2H, F, are not reliable and apparently presenting high background. Magnification of the images is required to show clear nuclear Slug expression as shown in Suppl Fig. 2d. The cell specific staining in several areas of Suppl Fig. S2f seems also non-specific (i.e., located outside ECs) and with high background. Specific Slug staining for CRC tumors, shown in Fig. 1, should be also included. Moreover, analyses of LacZ as reporter of the SlugKO allele is also required in the *in vivo* studies to unambiguously demonstrate Slug deletion in the corresponding tissues (tumor stroma and vessels; retina vessels).
4. Expression analysis shown in Fig. 2J are overexposed and difficult to interpret, please improve IFs.
5. Evidence to support the contribution of Slug to EndoMT is highly indirect and additional data are required. Expression of additional markers in EC-SlugOE and -siSlug cells at mRNA, WB and IF level is required. Moreover, functional analyses of required effectors of EndoMT (at least, MT1-MMP or MMP2 previously reported by the authors as mediating Slug effects, Welch-Reardon et al., 2014) are also needed, to confirm and support the gene expression data obtained by high throughput analyses (Fig. 4). Interpretation of GO analyses shown in Fig. 4A and B is very difficult

as the authors report at least two independent gene clusters (cluster 2 and cluster 6) containing EMT genes and both cluster exhibiting reverse expression patterns in Slug-OE cells (at 6 days) (Fig. 4A). These apparently contradictory findings require clarification of specific genes involved and functional studies.

6. Expression of the proposed EndoMT effectors in in vitro human ECs should also be analyzed in vivo: i.e., in CRC tumors and mouse retina. Otherwise, discussion on Slug and EndoMT is highly speculative.

7. The effect of Slug in proliferation of EC cells proposed and discussed by the authors is not apparently sustained by the in vivo data. No changes in proliferating cells (detected as Edu+) are observed in P5 retina samples of SlugKO compared to those of wt mice (Suppl Fig. S2a). These data are in clear contradiction to the proliferation observed (Edu+) in SlugOC EC cells compared to controls (Suppl Fig. S3b) and requires clarification.

8. Signaling analyses on ERK5 activation after SDF1 treatment (Fig. 7G, H) need to include detection of total ERK5. Also, analysis of ERK1/2 activation is required to support the conclusion included in Discussion section related to this aspect (page 21, first paragraph).

9. Is Dll4/Notch or VEGFR2 expression affected in CRC tumors obtained in control and Slug KO mice?

Minor points:

1. Data referring to melanoma B16 cells in SlugKO, described in page 4 (1st paragraph, last sentence) are not included in Suppl Fig S1a, or in other part of the ms. These data should be included or otherwise deleted from the text.

2. Description of data of Suppl. Fig. S1a and S1b, mentioned in page 9 (1st paragraph, line 5 and 2nd paragraph, line 8), is incorrect and appears to refer to Fig. S3a and S3b, respectively. Please correct.

Reviewer #2:

Remarks to the Author:

In the present study, Hultgren et al., studied the role of endothelial Slug in both developmental and pathological angiogenesis in context of tumor angiogenesis and retinal plexus development in mice. Authors revisited the phenotype of Slug KO mice (which have no obvious phenotype but attenuated inflammatory response) and show that Slug deficiency leads to reduced tumour angiogenesis and delayed (albeit transiently) retinal vascular development of both the primary and secondary vascular plexus. Furthermore, to unravel the underlying mechanisms by which endothelial Slug regulates vascular formation, the authors performed RNA-sequencing on Slug overexpressing endothelial cells (EC) harvested from an in vitro angiogenesis assay and identified that Slug expression promotes EC to lose junction proteins which is a hallmark of EndoMT. Moreover, via blocking VEGFR2 by Apatinib and Pazopanib, authors confirmed that Slug expression in EC promotes sprouting angiogenesis by modulating Notch signalling through repressing of Dll4 and induction of VEGFR2 expression, as has been previously published by the same authors. Finally, authors identified SDF1 α -CXCR4 pathway to stimulate angiogenesis by phosphorylation of ERK5 upon Slug activation.

The general topic is of relevance, the data appears sound and the manuscript is very well-written. However, this study appears to be a mere extension of previous studies, already published by the same group (Welch-Reardon et al., 2014). In this respect, the in vivo findings are new, while the majority of in vitro results are repeated from the previous study. In addition to this lack of novelty I have several other concerns:

Major points

1. To really prove that lack of endothelial Slug is responsible for the observed phenotype, an endothelial specific Slug knockout mouse should be studied.

2. No abstract is represented.

3. Unlike stated in the manuscript, no results for melanoma cells were represented (as indicated in the text for Fig.S1a).
4. Authors stated that no Slug+ astrocytes were found in the retinal plexus (Figure 2I). To provide evidence for this statement, double- staining for slug and an astrocyte marker has to be performed.
5. Unlike stated in the text on page 9 with respect to Slug overexpression, no data on proliferation is presented in the figure (Fig. S1b).
6. For the Apatinib and Pazopanib experiment, the important control of number of sprouts in Slug and GFP OE cells without drug treatment is missing (Fig. 5E-F).
7. In the ChIP-qPCR assay, authors intended to show Slug binding to the E-box sequences within the Dll4 promoter region. However, it is important to provide evidence that the +-500 bp region of DLL4 gene has a regulatory role.
8. The authors speculate that the reason why slug KO mice do not show any runt and vascular defects could be due to compensatory action of Snai1. If this is true, then why do the Slug KO mice show delayed retinal vascular development? This is an important point and needs to be addressed (e.g. by overexpression of Snai1 in slug KO mice, which should then have normal vascular development)
9. The authors remain unclear about the role of EndoMT in angiogenesis. The authors should take a position here.

Minor points:

1. In Fig.2H Immunofluorescence images for DAPI are over pixelated and in many figures scale bars are missing
2. Typos need to be fixed (e.g. on page 6, "shorty" should be "shortly", Fig. 3G labelling should be "Lacunarity" but not "Lancunarity, on page 20, "SDF SDF1 α -CXCR4" one SDF should be removed).
3. Figure labelling needs to be corrected (e.g. on page 11, Fig. S5d is not about junction protein expression levels, it should be S4d, on page12, Fig. 6D should be Fig. 5D)
4. In the sprouting assay in figure 3 and 7 lower magnification images should be included to give adequate overview.
5. In. Fig. 7J, the figure labelling "Pathological Angiogenesis" is partially invisible.

Reviewer #3:

Remarks to the Author:

The study by Hultgren et al. have addressed the role of Slug (Snail2) in endothelial cells, using a global Snail2 gene knockout mouse. Tumor growth and vascularization is suppressed in these mice and retinal vascular development is delayed. In Slug overexpressing HUVECs, the authors observe decreased expression of DLL4, Notch1 and NICD and dramatically increased expression of VEGFR2 and increased sprouting which is reversed by inhibition of VEGFR kinase activity. In the Slug KO mice, partial Notch inhibition rescues the retinal vascular outgrowth phenotype. The authors conclude their study by showing that SDF1a controls endothelial phenotype during angiogenic sprouting via Erk5. This is an ambitious study. It would have simplified interpretations if the authors had used endothelial cell-specific deletion of Slug.

1. Have the Slug KO endothelial cells in the retina vasculature lost their arterial/vein specification? In Fig. 2A, it looks as if this may be the case. If so, does it persist even when DLL4 expression is suppressed (see Fig. 6D). Endothelial cell proliferation occurs in the veins, not in the arteries. The authors state that there is no change in endothelial proliferation in the Slug KO retina (page 7 and Fig. S2a). Please assess proliferation in vein ECs specifically in the outer 20% of the retinal vascular front for each of the wt and Slug KO retinas and assess veins separately from arteries. In Fig. 2D, is vascular density assessed in the outer 20% of the vasculature or in the entire retinal leaflet? The authors state that it is migration, not proliferation, that is affected in Slug KO endothelial cells, however, they do not properly address the proliferation phenotype in the Slug

KO.

2. Were junctions in the Slug KO endothelial cells more enforced than in the WT? Was VEGF-induced vascular leakage affected? The authors have previously shown that in Slug siRNA-treated endothelial cells, VE-cadherin is not affected, however, in vitro cultures may not mimic the in vivo situation.

3. Were myeloid/hematopoietic differentiation affected in the Slug KO mice? It is well known that interference in these lineages affect endothelial development. Ideally, the authors should complement their study with a Slug iECKO mouse model.

4. Was Erk5 signaling in the retinal vascular front affected in the Slug KO mice at P6?

Additional comments

5. In the tumor studies, please inform on whether the tumor cells express Slug in vivo.

6. In Fig. 6B, it looks as if neuronal VEGFR2 expression is very considerably decreased in the Slug KO retinas. If so, this would affect the availability of VEGF. Clearly, the vascular phenotype in the Slug KO involve also non-endothelial cells.

Response to Reviewers

We thank the reviewers for their detailed and careful reviews. We address the comments of each reviewer individually and point-by-point. We note that there were numerous new and non-trivial experiments requested, many requiring the generation of new reagents and/or the development of new techniques. Where feasible, we have pursued additional studies to address these concerns, and of the several new experiments now included in our revision, particularly compelling, we believe, is the EC-specific rescue of Slug in the SlugKO mice, which restored retinal angiogenesis to close to WT levels. This confirms that Slug is required in EC for proper retinal angiogenesis. Thus, we have addressed all of the questions raised, and in aggregate our new data greatly strengthen the conclusion that Slug plays an important role in developmental angiogenesis and a critical role in pathological angiogenesis, and that it does so through regulation of notch signaling.

Our detailed responses (in black text) to reviewer comments (in grey text) are provided below.

Reviewer #1 (Remarks to the Author):

In the present ms. the authors analyze the involvement of Slug in angiogenesis, following their previous in vitro studies supporting a positive role of Slug (and Snail) in promoting angiogenesis in human endothelial cells (HUVEC) (Welch-Reardon et al., J Cell Sci, 2014). They now report the effect of Slug deletion in in vivo angiogenesis during retina development and tumor growth using a constitutive Slug KO model. Then, they moved to the in vitro system of HUVEC where they manipulate Slug expression (gain and loss of function) and analyze the effect at molecular and cellular level. The authors come to the conclusion that Slug controls angiogenesis by regulating the Dll4-Notch-VEGFR axis, as stated in the title. This study provides some information of interest that in fact complements other studies reporting a similar role, and mechanism of action, for Snail in angiogenesis (Wu et al., Nat Comms, 2014). Unfortunately, that previous finding decreases the general interest of the present study for a broad audience.

We thank Reviewer #1 for their careful reading of our manuscript, and we appreciate the observation that our findings are consistent with and complementary to previous studies, such as the cited study by Wu et al. (Nat Comm 2014). That group elegantly demonstrated that endothelial deletion of Snail1 (Snail) impairs embryonic vessel development due to dysregulation of Dll4/Notch signaling. However, we respectfully disagree that this and other earlier works reduces the general interest of our study. To date, the physiological role(s) and relative importance of Snail vs. Slug in embryonic vs. post-natal vessel remodeling has remained unclear, and has been the subject of significant controversy within the field. Indeed, although Wu et al. reported that embryonic deletion of Snail is lethal due to vessel formation defects, that group also

reported the rather unexpected observation that post-natal deletion of endothelial Snail fails to produce any notable defects in retinal angiogenesis. That group also did not characterize any effects of embryonic Snail deletion on residual endothelial Slug in their animals.

Our data help to resolve this conflict by presenting evidence that in contrast to embryonic development, in post-natal developing vessels endothelial-expressed Slug – not Snail – assumes an essential role in driving normal and pathological vessel growth and remodeling. We make extensive use of *in vivo* models of developmental (retina) and pathological (tumor xenograft) angiogenesis, as well as several *in vitro* models (sprouting angiogenesis bead assay, vascularized micro-organ platform), in combination with both loss-of-function and gain-of-function tools to manipulate Slug expression, to provide much-needed insight into the role of endothelial-expressed Slug in the post-natal context. Of general interest to readers is our finding that Slug regulates a partial endothelial-to-mesenchymal transition (which we extensively document using gene expression analysis and functional assays) to drive sprouting angiogenesis, and that loss of Slug expression leads to a failure of vessels to properly sprout, *in vitro* and *in vivo*, resulting in complete loss of tumor vascularization. Critically, we further find that endothelial re-expression of Slug rescues the retinal angiogenesis defect in these animals, but Snail is unable to compensate for the loss of Slug *in vitro* or *in vivo*. Taken together, these data demonstrate embryonic vs. post-natal vessel remodeling is distinctly regulated when it comes to Snail vs. Slug, and our studies specifically underscore Slug's fundamental role in supporting post-natal vessel growth and remodeling under normal and pathological conditions.

Thus, we believe our data are of significant general interest to readers because they provide novel insight into the signaling events that govern sprouting angiogenesis, which we show requires a partial EndoMT event driven by Slug-mediated Notch signaling in post-natal blood vessels. Our data also help resolve current controversy in the field over the relative roles of Snail vs. Slug by demonstrating that Slug seems to predominate in the regulation of post-natal blood vessels, while Snail may perform this function primarily in the embryonic context. Future work – understandably beyond the scope of this already extensive study -- will be undertaken to better understand how and why endothelial Slug takes over for Snail in supporting this function in post-natal vessels.

In addition, there are several important concerns with the *in vivo* and *in vitro* data that require further clarification and additional experimentation. In particular, the proposed implication of Slug in mediating a partial endothelial-mesenchymal transition (EndoMT) is not convincingly demonstrated in the *in vivo* Slug KO model. Other concern refers to what extent all *in vitro* data, including the proposed mechanism for Slug in angiogenesis obtained from human EC, can be extrapolated to the *in vivo* mouse model.

We respectfully disagree with this assessment of our *in vivo* data, which we argue convincingly demonstrates a partial EndoMT process when considered in conjunction with our extensive *in vitro* studies. Our finding is that a known EMT regulator, Slug, is upregulated during angiogenesis, and that it drives the expression of several well-recognized “EMT genes”. However, as the endothelial cells do not fully separate from each other, even though junctional proteins are down-regulated, we feel that the process is best-described as a “partial EndoMT”. Specifically, we show in the SlugKO animal that both tumor angiogenesis and retinal vessel outgrowth are significantly impaired, the latter in association with increased tip cell number and decreased EC proliferation, suggesting a defect that delays sprout invasion of avascular tissue. New experiments included in this revision further show that endothelial-specific re-expression of Slug rescues this phenotype. To understand the mechanism underlying this phenotype, we performed exhaustive *in vitro* studies in several sophisticated systems involving human-derived primary cells that we have routinely found faithfully reflect the processes occurring *in vivo*, while also providing strong evidence of relevance to the human context. Indeed, the bead sprouting assay in particular is a well-established *in vitro* model of sprouting angiogenesis that is in use by ourselves as well as many other labs who also study this process. Several published studies from our lab and others repeatedly demonstrate the ability of this model to faithfully mirror physiological responses that occur *in vivo* in the mouse – and, importantly also, the human -- setting.

In the *in vitro* models we use in the current studies, we have performed extensive gene expression analysis as well as functional assays to demonstrate that normal EC proliferation, migration, and junctional stability – well-established hallmarks of EndoMT – are all dependent upon Slug. New studies included in this revision further translate our *in vitro* studies back to the mouse by demonstrating that VEGF-induced vessel permeability is markedly impaired in SlugKO animals, which correlates with the loss of junctional proteins when Slug expression is upregulated. When taken together, our *in vitro* and *in vivo* work is consistent with earlier reports by other groups that have already documented the role of Slug in EMT, and that have already helped establish the concept of full EndoMT as well as partial EMT. Importantly, however, the idea of a Slug-driven partial EndoMT driving post-natal vessel growth remains the novel finding in this manuscript that should be of broad general interest to readers.

The ms. also suffer from interpretation of data (in particular gene expression studies) and contains some formal defects. Overall, in my opinion, the ms. does not provide convincing evidence to support the main authors’ hypothesis and only provides incremental knowledge on the implication of EMT factors of the Snail family in angiogenesis.

This manuscript has undergone significant revision and includes several new studies as requested by reviewers. The outcomes of these studies – including a significantly expanded analysis of our gene expression studies (discussed in greater detail below) – provide even more evidence in support of

our interpretation of the data. We regret that our original submission contained some typographical errors, which we have since corrected in this resubmission. We respectfully disagree with Reviewer #1's overall assessment of this manuscript, and hope that our point-by-point responses below alleviates those concerns.

Main points:

1. Tumor data in Slug KO model. The authors analyze the tumorigenic capacity of two syngeneic colorectal cell lines in wt and Slug KO mice. Those cell lines should be described and provide references. Is the proliferation or apoptosis of tumors affected in the absence of Slug? This is an important point requiring clarification in view of the prosurvival action of Slug in other tumor contexts. These studies should be complemented with additional tumor models (like syngeneic breast carcinoma and using orthotopic injections) to prove the generality of the observations.

We have now added references for each cell line (Refs 31,32). We have also included the data we generated with syngeneic melanoma B16 cells along with a reference to these cells (Suppl. Fig. S1c, Ref 33). The reviewer also requested clarification on how loss of Slug affects tumor apoptosis and proliferation as Slug is known for its pro-survival effect in tumors. We feel that the reviewer may have misunderstood the system we utilized here as the tumor cells themselves are not known to be Slug-deficient. We implanted "wild type" tumor cells into either WT or SlugKO mice to demonstrate that the absence of Slug in the host animal's endothelial cells specifically negatively affected tumor angiogenesis despite "normal" Slug expression in the tumor cells themselves. This lack of tumor angiogenesis in turn drastically reduced tumor growth, again, despite "normal" Slug expression in the tumor cells. Any effects on tumor cells are thus a result of the environment in which they find themselves. A comprehensive study of these changes within the tumor cells is, we believe, outside the scope of this manuscript.

2. Moreover, the authors should comment on other studies on in vivo tumorigenesis in the same Slug KO mice model, as in skin tumorigenesis induced by UV-radiation (Newkirk et al., Am J Pathol 171: 1629-39, 2007) or chemical carcinogenesis (Villarejo et al., Carcinogenesis 36: 585-97, 2015), with different outcomes. The later study shows that in fact Slug deletion caused increased tumor progression by inducing a pro-inflammatory response. The authors should therefore discuss those data in the context of the present study.

We have now addressed these concerns in the Discussion section. Briefly, both the Newkirk study and the Villarejo study used UV or chemically-induced tumor models to promote tumorigenesis in Slug-deficient animals, resulting in tumor cells lacking normal Slug expression. Resulting immune activation observed in these models may be specific to that type of challenge. By contrast,

we utilized a sub-cutaneous tumor implantation model to introduce “wild-type” tumor cells with unmanipulated Slug expression. Moreover, the Villarejo study used the LacZ strain of SlugKO mice, which only has exons 2-3 deleted and replaced with LacZ. These animals were then backcrossed to Friend leukemia virus-B type mice. We used a different strain from Jackson Labs (B6:129S1-Snai2^{tm2Grid}/J), which has complete deletion of Slug. It is also on a B6/C57-129S background. Thus the differences in genetic background might also have contributed to differences in inflammatory responses. Interestingly, a recent study (El-Brolosy, et al. Nature568: 193-197, 2019) has shown that genetic compensation can be triggered by decaying mutant RNA present in the mutant animals – a result of the genetic mutation only truncating the transcript. This is not seen where the entire transcript is lost, as in our mice. Thus, truncated Slug mRNA in the LacZ strain could have triggered compensatory upregulation of related genes such as Snail. Finally, a recent publication from the same group as the Newkirk study suggests that Slug and Snail expression actually promote hyperplasia and inflammation in UVR and chemically-induced skin cancer (Ref 65), which reconciles nicely with our finding that Slug-deficiency leads to reduced tumor growth.

3. Slug expression in in vivo samples (tumors, retina) needs improvement. IF data show in Fig. 2H, F, are not reliable and apparently presenting high background. Magnification of the images is required to show clear nuclear Slug expression as shown in Suppl Fig. 2d. The cell specific staining in several areas of Suppl Fig. S2f seems also non-specific (i.e., located outside ECs) and with high background. Specific Slug staining for CRC tumors, shown in Fig. 1, should be also included. Moreover, analyses of LacZ as reporter of the SlugKO allele is also required in the in vivo studies to unambiguously demonstrate Slug deletion in the corresponding tissues (tumor stroma and vessels; retina vessels).

We have now included new images to demonstrate endothelial nuclear expression of Slug (Fig. 2I). It is worth noting that due to the limitation of antibody quality, and relatively low expression of Slug as a transcription factor, it has been difficult for us to obtain images with absolutely no background. Nonetheless, we have further optimized our staining and imaging to further reduce background fluorescence. The reviewer also asked for Slug staining in CMT93 and MC38 tumors, and LacZ staining showing absence of Slug in tumor stroma. However, it has been shown by other studies that Slug is expressed in CMT93 and MC38 tumor cells (Ref 31, 32), and as noted above in our studies, these “wild-type” (i.e. Slug-expressing) cells implanted into a Slug-null host background.

As for LacZ staining, the SlugKO mouse strain (B6:129S1-Snai2^{tm2Grid}/J) we used does not have the LacZ expression cassette (which is present for the other SlugKO strain available from the Jackson Lab B6.129S1-Snai2^{tm1Grid}/J) so we would not be able to definitively demonstrate loss of Slug in tumor stroma. However, we believe that the absence of Slug in the SlugKO retina vessels compared to the WT mice as seen by immunofluorescent staining (Figure 2I, Suppl Fig.2d) is sufficient proof for Slug knockout in EC.

4. Expression analysis shown in Fig. 2J are overexposed and difficult to interpret, please improve IFs.

We have included new images with better resolution and improved exposure (Suppl Fig. S2f).

5. Evidence to support the contribution of Slug to EndoMT is highly indirect and additional data are required. Expression of additional markers in EC-SlugOE and –siSlug cells at mRNA, WB and IF level is required. Moreover, functional analyses of required effectors of EndoMT (at least, MT1-MMP or MMP2 previously reported by the authors as mediating Slug effects, Welch-Reardon et al., 2014) are also needed, to confirm and support the gene expression data obtained by high throughput analyses (Fig. 4). Interpretation of GO analyses shown in Fig. 4A and B is very difficult as the authors report at least two independent gene clusters (cluster 2 and cluster 6) containing EMT genes and both cluster exhibiting reverse expression patterns in Slug-OE cells (at 6 days) (Fig. 4A). These apparently contradictory findings require clarification of specific genes involved and functional studies.

To address the need for additional proof of EndoMT, we have now included a EMT-signature gene heatmap, mRNA, and WB data on additional EndoMT markers such as alphaSMA, SPARC, etc. (Suppl Fig. S4b, S4c, S4d), One of the clearest hallmarks of an EMT/EndoMT phenotype is changes in cell junction/barrier function. In the original manuscript, using our *in vitro* 3D vascularized micro-organ model, we demonstrated that SlugOE vasculature has severely impaired barrier function due to loss of various EC junctions (Fig. 4F). To confirm this result *in vivo*, we have now included an *in vivo* VEGF-induced permeability experiment showing that the reverse situation is also true. As expected, SlugKO mouse vessels are less permeable compared to WT (Suppl. Fig. S4h). In addition, the original manuscript showed that SlugOE EC have increased proliferation rates (Suppl. Fig. S3b), also characteristic of EMT, and this is supported by gene expression data (Fig. 4A, 4B, 4C, Suppl. Fig. S4b, S4c, S4d, S4f). We completely agree that another feature of EMT/EndoMT is increased cell migration and MMP activity as mentioned by the reviewer. In fact, we did find increased pro-migratory gene expression in our RNA-seq data (Fig. 4A, 4B). However, we feel that the functional analysis of the MMPs in Slug overexpressing EC have been reported in detail by our previous study (Welch-Reardon et al., 2014) as the reviewer also pointed out, and therefore not within the scope of this investigation.

We agree with the reviewer that gene expression data and GO analyses are often hard to interpret. We have now included an additional graph (Fig. 4B) showing GO categories in each of the 10 clusters, demonstrating that the downregulated clusters mainly include genes involved in cell junctions, adhesion, ECM, cytoskeleton arrangement, cell polarity, as well as changes in cell shape and ECM components. The downregulated clusters also included more negative

regulators of proliferation and migration, which would translate into a net positive effect on proliferation and migration. Specifically, clusters 3 and 5 are the most strongly associated with EMT (Fig. 4C). Cluster 3 contains many down-regulated junctional genes, while cluster 5 shows strong up-regulation of proliferation and migration at day 6. Proliferation is also strongly associated with cluster 7, but at day 2. So, cluster 7 may represent genes involved early in proliferation, while cluster 5 is presumably associated with genes controlling proliferation at later times. We also see down-regulation of junctional genes in cluster 1, as well as down-regulation of negative-regulators of EMT, such as FoxP4 (down 11-fold) and HHIP (down 3.5-fold). We do not see the data as contradictory, rather they are an indication that there is a lot more to understand about the complex interrelationships between genes during cell proliferation and morphogenesis.

6. Expression of the proposed EndoMT effectors in *in vitro* human ECs should also be analyzed *in vivo*: i.e., in CRC tumors and mouse retina. Otherwise, discussion on Slug and EndoMT is highly speculative.

The reviewer's concern is understood, however, we have consistently found that the data we generate in our sophisticated *in vitro* systems faithfully mirror what we see *in vivo*, and thus we do not see our conclusions as being "highly speculative". Indeed, the role of Slug in EMT is well known, the process of EndoMT is well-described, and the concept of partial EMT's is also widely accepted. We use *in vitro* models to dive into mechanism in a way that is hard to do using mice. Importantly, however, in this paper we have shown that our *in vitro* findings of a role for Slug in vessel formation, and the critical part that notch signaling plays in that, both translate perfectly to the mouse. We have used *in vitro* models to then probe changes in gene expression and find that these are consistent with numerous previously published reports on the role of Slug in EMT. Our argument here then is that our data are consistent with a similar role for Slug in EC, with the caveat that since we do not see complete dissolution of EC-EC junctions, the changes are more consistent with a partial endoMT. The process is presumably limited by associated regulatory gene networks, and we have preliminary data hinting at what these might involve.

7. The effect of Slug in proliferation of EC cells proposed and discussed by the authors is not apparently sustained by the *in vivo* data. No changes in proliferating cells (detected as Edu+) are observed in P5 retina samples of SlugKO compared to those of wt mice (Suppl Fig. S2a). These data are in clear contradiction to the proliferation observed (Edu+) in SlugOC EC cells compared to controls (Suppl Fig. S3b) and requires clarification.

We want to thank the reviewer for noting this inconsistency in our data. Upon discussion with experts in retinal vessel morphometry, we learned that differences in EC proliferation can be difficult to broadly detect in this tissue without more careful analysis, even in knockout mice that display profound and well-established defects in EC retinal angiogenesis and sprouting. We have now

repeated the *in vivo* EdU incorporation experiment in mouse retina and adopted an improved and more thorough analysis protocol. We now report a reduction in EdU⁺ cells in both arterial and venous plexi in P6 SlugKO mouse retina compared to the control (Fig. 2H), which is consistent with our *in vitro* finding that SlugOE EC show increased EdU incorporation (Suppl. Fig. S3d).

8. Signaling analyses on ERK5 activation after SDF1 treatment (Fig. 7G, H) need to include detection of total ERK5. Also, analysis of ERK1/2 activation is required to support the conclusion included in Discussion section related to this aspect (page 21, first paragraph).

We have now included the requested data (Fig. 7G, 7H). The reviewer also mentions the analysis of ERK1/2 in the context of our conclusion in the Discussion section. Upon further investigation, we realize that we cannot rule out the involvement of ERK1/2 in this process and so have now removed this conclusion from the discussion. We apologize for the mistake.

9. Is Dll4/Notch or VEGFR2 expression affected in CRC tumors obtained in control and Slug KO mice?

We thank the reviewer for this excellent suggestion, however, given that we show that Slug clearly plays a role in both post-natal developmental and pathologic angiogenesis, and given the extensive analysis we have already presented on notch in the retina, we feel that further investigation of this pathway in tumors is probably beyond the scope of this already quite extensive study.

Minor points:

1. Data referring to melanoma B16 cells in SlugKO, described in page 4 (1st paragraph, last sentence) are not included in Suppl Fig S1a, or in other part of the ms. These data should be included or otherwise deleted from the text.

We have included the previously omitted data (Suppl. Fig. S1e) and apologize for the error.

2. Description of data of Suppl. Fig. S1a and S1b, mentioned in page 9 (1st paragraph, line 5 and 2nd paragraph, line 8), is incorrect and appears to refer to Fig. S3a and S3b, respectively. Please correct.

We have corrected the in-text reference to the figures and apologize for the omission.

Reviewer #2 (Remarks to the Author):

In the present study, Hultgren et al., studied the role of endothelial Slug in both developmental and pathological angiogenesis in context of tumor angiogenesis and retinal plexus development in mice. Authors revisited the phenotype of Slug KO mice (which have no obvious phenotype but attenuated inflammatory response) and show that Slug deficiency leads to reduced tumour angiogenesis and delayed (albeit transiently) retinal vascular development of both the primary and secondary vascular plexus. Furthermore, to unravel the underlying mechanisms by which endothelial Slug regulates vascular formation, the authors performed RNA-sequencing on Slug overexpressing endothelial cells (EC) harvested from an *in vitro* angiogenesis assay and identified that Slug expression promotes EC to lose junction proteins which is a hallmark of EndoMT. Moreover, via blocking VEGFR2 by Apatinib and Pazopanib, authors confirmed that Slug expression in EC promotes sprouting angiogenesis by modulating Notch signalling through repressing of Dll4 and induction of VEGFR2 expression, as has been previously published by the same authors. Finally, authors identified SDF1 α -CXCR4 pathway to stimulate angiogenesis by phosphorylation of ERK5 upon Slug activation.

The general topic is of relevance, the data appears sound and the manuscript is very well-written. However, this study appears to be a mere extension of previous studies, already published by the same group (Welch-Reardon et al., 2014). In this respect, the *in vivo* findings are new, while the majority of *in vitro* results are repeated from the previous study.

We thank Reviewer #2 for their thoughtful review of this manuscript, and agree that the topic is of general relevance and interest. Furthermore, we feel that this manuscript provides substantial and novel insight into the role of Slug in sprouting angiogenesis. Although our earlier paper introduced the concept that Slug regulates sprouting angiogenesis by an EMT-like process, the current study significantly expands upon this idea. We use both loss-of-function and gain-of-function tools to both eliminate and rescue Slug -- in both *in vivo* and *in vitro* models -- to comprehensively document Slug's role in governing partial EndoMT in sprouting angiogenesis. In contrast to our earlier study, we perform extensive gene expression analysis to document Slug's effect on hallmark processes of EMT/EndoMT, and we support those observations with functional studies conducted both *in vitro* and *in vivo*. Thus, we respectfully disagree with the assessment that this study is a "mere extension" of earlier work. Rather, we believe that the earlier study laid the initial groundwork and rationale for this far more extensive and comprehensive study that now provides several novel and key pieces of evidence regarding Slug's role in mediating a partial EndoMT to support sprouting angiogenesis in post-natal physiological and pathological settings of vessel growth. We further hope that our point-by-point address to Reviewer #2's specific concerns, provided below, will help alleviate the reviewer's concerns.

In addition to this lack of novelty I have several other concerns:

Major points

1. To really prove that lack of endothelial Slug is responsible for the observed phenotype, an endothelial specific Slug knockout mouse should be studied.

That is certainly one approach, however, we are interested in not only the endothelial cells but also in how they interact with their local microenvironment. The use of a global knockout with cell-specific rescue is an ideal way to investigate these interactions. A key question the reviewer might be alluding to here is whether Slug is really needed in the EC. We are confident the answer is yes for the following reasons: 1) multiple *in vitro* assays using knockdown and overexpression clearly show that Slug is required for sprouting angiogenesis, a process that is remarkably similar in our *in vitro* assays to the process *in vivo*; 2) other cells proximal to the EC *in vivo* either do not express Slug (astrocytes) or show no obvious changes when Slug is absent (pericytes); and, 3) perhaps most definitively, EC-specific rescue of Slug expression restores retinal vasculature in Slug knockout mice. Specifically, via EC-specific re-expression of Slug on a VE-cad Cre+, SlugKO mouse background we demonstrated that lentiviral-delivered, VE-Cre mediated EC-specific restoration of mSlug leads to increased vascular density compared to the GFP-control expressing group (Fig. 2J,K, Suppl Fig. S2n, o). We believe our studies are sufficient to support our contention that endothelial Slug expression has an important role in vascular development/angiogenesis.

2. No abstract is represented.

We believe there was an uploading error. We apologize, and have now included the abstract with the main text.

3. Unlike stated in the manuscript, no results for melanoma cells were represented (as indicated in the text for Fig.S1a).

We apologize that the melanoma data referenced in the text were accidentally omitted. These data are now included (Suppl Fig. S1e).

4. Authors stated that no Slug+ astrocytes were found in the retinal plexus (Figure 2I). To provide evidence for this statement, double- staining for slug and an astrocyte marker has to be performed.

An IF image was actually included in the original manuscript (Fig. 2J of the original figure, now Fig. S2e), however, we have now included new images with higher magnification (Fig. 2I) to better illustrate this point.

5. Unlike stated in the text on page 9 with respect to Slug overexpression, no data on proliferation is presented in the figure (Fig. S1b).

We apologize for the typo. We have now corrected the in-text reference (Suppl. Fig. S3d).

6. For the Apatinib and Pazopanib experiment, the important control of number of sprouts in Slug and GFP OE cells without drug treatment is missing (Fig. 5E-F).

This was actually included in the original manuscript (Fig. 5E-F, first and second bar group from the left)

7. In the ChIP-qPCR assay, authors intended to show Slug binding to the E-box sequences within the Dll4 promoter region. However, it is important to provide evidence that the +/-500 bp region of DLL4 gene has a regulatory role.

This is a good point. Others have previously shown using ChIP-seq that the same region of the Dll4 promoter is bound by Snail, also through these E-boxes. We consider ChIP-seq data as being an excellent indication that these are indeed active sites and have now included the reference (Ref 55). We have also noted a numbering error when describing the location of these sites and this has also been fixed.

8. The authors speculate that the reason why slug KO mice do not show any runt and vascular defects could be due to compensatory action of Snai1. If this is true, then why do the Slug KO mice show delayed retinal vascular development? This is an important point and needs to be addressed (e.g. by overexpression of Snai1 in slug KO mice, which should then have normal vascular development)

9. The authors remain unclear about the role of EndoMT in angiogenesis. The authors should take a position here.

To be clear, we do see some runting in the SlugKO mice, as have others, and we do see defective vessel formation in the retina and in tumors. The question is why is the SlugKO not lethal, and here we do believe that the presence of Snai1 and/or related transcription factor may be providing partial, but nonetheless incomplete, compensation for ablation of Slug during development. Furthermore, earlier groups reported that post-natal deletion of Snail in EC produced no retinal angiogenesis phenotype, whereas we observe a striking retinal vasculature defect in SlugKO animals along with a profound failure of these animals to support tumor angiogenesis. Thus, SlugKO animals may not die embryonically because while Snail is essential for embryonic vessel development, Slug may assume greater importance in post-natal vessels compared to embryonically. Consistent with this interpretation, we did perform qPCR analysis on sorted retinal EC from WT and SlugKO and found, perhaps surprisingly, that Snai1 expression was not increased (as it is during embryonic

heart valve formation and chondrogenesis, Ref 25, 41). It is increasingly recognized that Snai1, Snai2 (Slug), Twist, and the numerous other transcription factors that regulate EMT do not really act in a linear pathway, but rather operate in a series of feedback loops. As noted above, we already have some evidence of a family of transcription factors in EC that might “apply the brakes” to the EMT process, giving rise to the partial EndoMT we see during angiogenesis. It is possible that this mechanism also compensates to some extent for the loss of Slug. We have now included the new data on Snai1 (Suppl Fig.2i) and have made some changes to the Discussion to address these possible mechanisms.

Minor points:

1. In Fig.2H Immunofluorescence images for DAPI are over pixelated and in many figures scale bars are missing

We have updated the image (now Fig. 2I) and included all the scale bars.

2. Typos need to be fixed (e.g. on page 6, “shorty” should be “shortly”, Fig. 3G labelling should be “Lacunarity” but not “Lancunarity, on page 20, “SDF SDF1 α -CXCR4” one SDF should be removed).

We have fixed the typos.

3. Figure labelling needs to be corrected (e.g. on page 11, Fig. S5d is not about junction protein expression levels, it should be S4d, on page12, Fig. 6D should be Fig. 5D)

We have corrected this labeling.

4. In the sprouting assay in figure 3 and 7 lower magnification images should be included to give adequate overview.

We have included the lower magnification images (Fig. S3b, S6a).

5. In. Fig. 7J, the figure labelling “Pathological Angiogenesis” is partially invisible.

We have updated the figure.

Reviewer #3 (Remarks to the Author):

The study by Hultgren et al. have addressed the role of Slug (Snail2) in endothelial cells, using a global Snail2 gene knockout mouse. Tumor growth and vascularization is suppressed in these mice and retinal vascular development is delayed. In Slug overexpressing HUVECs, the authors observe decreased expression of DLL4, Notch1 and NICD and dramatically increased expression of VEGFR2 and increased sprouting which is reversed by inhibition of VEGFR kinase activity. In the Slug KO mice, partial Notch inhibition rescues the retinal vascular outgrowth phenotype. The authors conclude their study by showing that SDF1a controls endothelial phenotype during angiogenic sprouting via Erk5. This is an ambitious study. It would have simplified interpretations if the authors had used endothelial cell-specific deletion of Slug.

We thank Reviewer #3 for this review. Although endothelial-specific Slug deletion was one way to address the role of Slug in retinal angiogenesis, we feel that our approach using global Slug knockout with EC-specific rescue would allow us to identify both EC-specific effects of Slug as well as other effects that might contribute to sprouting angiogenesis from a Slug-null microenvironment. Importantly, we find that despite the profound effect of Slug deletion on post-natal retinal angiogenesis and tumor angiogenesis, EC-specific re-expression of Slug almost completely rescues the retinal phenotype. This underscores the importance of endothelial-expressed Slug in this process, while providing more information than would have been obtained had we pursued this question using only EC-specific SlugKO animals.

1. Have the Slug KO endothelial cells in the retina vasculature lost their arterial/vein specification? In Fig. 2A, it looks as if this may be the case. If so, does it persist even when DLL4 expression is suppressed (see Fig. 6D). Endothelial cell proliferation occurs in the veins, not in the arteries. The authors state that there is no change in endothelial proliferation in the Slug KO retina (page 7 and Fig. S2a). Please assess proliferation in vein ECs specifically in the outer 20% of the retinal vascular front for each of the wt and Slug KO retinas and assess veins separately from arteries. In Fig. 2D, is vascular density assessed in the outer 20% of the vasculature or in the entire retinal leaflet? The authors state that it is migration, not proliferation, that is affected in Slug KO endothelial cells, however, they do not properly address the proliferation phenotype in the Slug KO.

To address these concerns we have conducted α SMA staining in WT and SlugKO mouse retina at P5/6 and quantified the level of arterialization by measuring the longest α SMA+ vessel length as a percentage of the total length of the same branch. We observed no significant differences between WT and SlugKO retina (Suppl. Fig. S2b) and therefore we conclude that there are no obvious arterial/venous specification defects in SlugKO mice in the context of this study.

The reviewer also asked us to provide more specific and accurate assessments of cell proliferation in SlugKO mouse retinas. We have now repeated our *in vivo* EdU experiment and adopted better and more thorough analysis methods to assess arterial and venous regions separately. Based on our improved quantification we now report a reduction in EdU incorporation in both arterial and venous regions of the SlugKO retinal primary plexus at P5/6, suggesting that Slug-deficiency leads to decreased EC proliferation. We thank the reviewer for this suggestion.

2. Were junctions in the Slug KO endothelial cells more enforced than in the WT? Was VEGF-induced vascular leakage affected? The authors have previously shown that in Slug siRNA-treated endothelial cells, VE-cadherin is not affected, however, *in vitro* cultures may not mimic the *in vivo* situation.

This is an interesting point. We have now included an *in vivo* VEGF-induced permeability experiment, the Miles Assay. We found that SlugKO mice indeed had better barrier functions and were less susceptible to VEGF-induced hyper-permeability (Suppl. Fig. S4g). This may reflect lower VEGFR2 expression and therefore reduced responsiveness to VEGF induction.

3. Were myeloid/hematopoietic differentiation affected in the Slug KO mice? It is well known that interference in these lineages affect endothelial development. Ideally, the authors should complement their study with a Slug iECKO mouse model.

It is certainly true that myeloid lineage cells contribute to angiogenesis, and the lack of myeloid-expressed Slug in our animals might contribute to the defects. However, we expect that any effects are probably small at best, for the following reasons: 1) multiple *in vitro* assays using knockdown and overexpression in EC clearly show that Slug is required in EC for sprouting angiogenesis, a process that is remarkably similar in our *in vitro* assays to the process *in vivo*; and, 2) perhaps most definitively, EC-specific rescue of Slug expression restores retinal vasculature in Slug knockout mice. Specifically, via EC-specific re-expression of Slug on a VE-cad Cre+, SlugKO mouse background we demonstrated that lentiviral-delivered, VE-Cre mediated EC-specific restoration of mSlug leads to increased vascular density compared to the GFP-control expressing group (Fig. 2J, K, Suppl Fig. S2n, o). Thus we believe that it is the loss of Slug in the EC that is the major contributor to the vascular defects we see in these mice.

4. Was Erk5 signaling in the retinal vascular front affected in the Slug KO mice at P6?

It is an interesting question and is definitely worth further investigating. However, due to limitations of IF antibody available for pERK5, it is hard to directly assess ERK5 signaling activities in SlugKO mouse retina. In our study,

we found that SDF1 α -CXCR4 activation of pERK5 leads to Slug induction and therefore sprouting. It is known that SDF1 α -CXCR4 signaling is important for tip cell behavior and vascular patterning, therefore we speculate that it is possible ERK5 signaling is defective in SlugKO mouse retinal vascular front as well. Hopefully, improved reagents will allow direct testing of this idea in the future.

Additional comments

5. In the tumor studies, please inform on whether the tumor cells express Slug in vivo.

As noted above, these are wild-type tumors cells being transplanted into SlugKO mice and so, based on previous studies showing Slug expression in both CMT-93 and MC-38 tumor cells (Ref 31, 32), we would expect them to express Slug in our assays.

6. In Fig. 6B, it looks as if neuronal VEGFR2 expression is very considerably decreased in the Slug KO retinas. If so, this would affect the availability of VEGF. Clearly, the vascular phenotype in the Slug KO involve also non-endothelial cells.

We thank Reviewer #3 for making this point. However, we note that if neuronal VEGFR2 is also decreased, it would likely make *more* VEGF available for EC, potentially *improving* their sprouting. Moreover, as discussed above, the EC-specific Slug rescue experiment on the SlugKO background provides compelling evidence for a critical role of EC-expressed Slug in regulating angiogenesis.

Reviewers' Comments:

Reviewer #1:

Remarks to the Author:

This is an improved version of the original ms. by Hultgren et al. describing Slug involvement in the control of EndoMT and in angiogenesis. In the present revised version the authors have addressed most of the concerns of this reviewer by performing additional experiments and/or providing solid arguments. They also provide additional evidence for the participation of Slug in in vivo angiogenesis by performing elegant Slug-EC rescue experiments in Slug KO mice. While most of the previous queries have been properly answered there are still some concerns that have not been fully addressed requiring additional elaboration. Besides, there are some additional points in the present revised version that require clarification before publication

Overall appraisal of the revised version:

My previous concern on the lack of novelty of the present work based on the previous in vitro study of the authors (Welch-Reardon et al., 2014) and the previous work of other group reporting a similar role for Snail in in vivo angiogenesis (Wu et al., Nat Comms, 2014), has been properly addressed by the authors in the rebuttal letter and additional support from the revised ms. Similarly, the authors have provided stronger evidence and arguments to support the proposal that Slug mediates a partial EndoMT process in EC in vivo and in vitro. The doubts on the analyses of RNA-seq data are also much clarified. I am now more convinced on the interest and relevance of the present study regarding Slug involvement in physiological angiogenesis, but its relevance for tumor angiogenesis is still not completely clear.

Main points:

Related to the original review:

1. Tumor data in Slug KO model.

I appreciate and thank the authors for the detailed information and clarification of the specific Slug KO model used here, and the differences with Slug KO models used in other studies that were not evident to this reviewer in the original ms. Also, I appreciate the clarification regarding the experimental tumor model of xenografting "wild type" CRC cells into either wild type or constitutive Slug KO mice. They have also included some of the requested information on the mouse CRC cell lines, including appropriate references (#32,33) (please, note that ref. 31 is not appropriate to this point). However, the authors still do not indicate the differences between the two CRC cell lines used, that are described in the original references: CMT93 (epithelial and low tumorigenic cells) and MC38 (mesenchymal and highly tumorigenic cells) (#32 y 33) that can be important to consider here.

In response to a related question on the cell lines raised by reviewer #3 on endogenous Slug expression, the authors argue that "based on previous studies showing Slug expression in both CMT-93 and MC-38 tumor cells (Ref 31, 32), we would expect them to express Slug in our assays". I would like to remark that Slug expression in refs 32, 33 (not 31 and 32) is analyzed by RT-PCR with apparent low or very low Slug mRNA levels in MC38 cells (33). I still think it is important to know whether tumor cells express or not endogenous Slug to correctly interpret the results of tumor xenografts and, thus, the authors should confirm Slug expression in the two CRC cell lines at protein level, by WB.

In addition, they need to explain the apparent contradictory data on tumor growth observed in MC38 cells at end points presented in Fig. 1B (no apparent tumor growth at end point) and Suppl Fig. S1a (left, median tumor growth around 500 mm³). Indeed, if the latter data are correct, MC38 tumors should be analyzed for angiogenesis and necrosis, instead of CMT93 tumors presented in Suppl Fig. S1b.

Moreover, I still think that analyses of the proliferative status of the xenografted tumors in wild type and Slug KO mice is strictly required to complement the analysis of angiogenesis and necrosis

(Supp Fig. S1b). This can be easily done by IHQ analysis of Ki67 or other proliferation marker in tumor sections. Otherwise to assign the observed differences only to angiogenic defects is not fully sustained.

2. Reference to other tumor studies in Slug KO mice.

The authors have included this information in the Discussion section as requested (page 20, lines 3-6). However, they should realize that the cited ref#67 (Villarejo et al., 2015) report on increased (instead of reduced) tumor growth in Slug KO mice, as indicated in the article title. Therefore, these differences should be properly commented.

Additional queries on the revised version:

1. ChIP analysis of Slug binding to Dll4 promoter (Fig 5D) is highly confusing. The authors indicate that they analyze the first E-box sequence contained 140 bp downstream of the Dll4 TSS, as referred in ref. 55 (Wu et al., 2014), using primers located between around +100 and +200 bp (information of the precise localization of the primers is not provided). However, the information reported by Wu et al. indicates that the proximal murine Dll4 promoter contains three E-boxes (-432/-427; -395/-390; and +43/+48) functionally bound for Snail (please, see Fig. 7c-e in Wu et al.). According to this information, no E-box element should be present in the amplified region bound by Slug in the Dll4 promoter analyzed here (Fig. 5D). In the rebuttal letter, the authors explain a previous mistake in the numbering of the Dll4 promoter region presented in the original ms; however, the information presented in the revised version is still incorrect or at least not coinciding with the mentioned previous work. Clarification of this point is important, in particular considering that in fact the authors do not provide direct evidence for a functional action of Slug on the Dll4 promoter. Promoter assays in EC cells with wt and mutant E-boxes in the presence and absence of Slug (similar to those performed in Wu et al. for Snail) should be included in the present ms. to unambiguously support the proposed repression action of Slug on Dll4 transcription, as well as on the Jag2 promoter. Otherwise, only Slug binding to those promoter regions can be claimed but not functional transcriptional effects.
2. Knockdown analyses of CXCR4 and CXCR7 (Fig. 7H, S6h) lack proper controls to show efficient knockdown of the corresponding genes (i.e., decreased endogenous CXCR4 and CXCR7 mRNA levels).
3. IF images presented in Fig. S4g are of low quality (high background and apparent non-specific stain). Please, improve them or delete this specific information.
4. Western blots presented in Fig. 5C and S5a, lack lane labeling and loading controls (except for Dll4 expression).
5. In Fig. 1D, please include magnification areas of matrigel plugs of Slug KO mice to more faithfully compare with the plugs of WT mice.
6. Data of lentiviral Slug overexpression in EC, shown in Fig. 3A and S3c, appear to this reviewer to be similar, or the same, as the data presented in the previous work of the group (Fig. 3 in Welch-Reardon et al., 2014). Please, clarify the differences.
7. Page 11, 2nd paragraph: Conclusion on Slug role in developmental angiogenesis. Please, change "indispensable" for important.
8. Page 16, 2nd paragraph, lines 3-6. Please note that information presented in Fig S5a refers to Slug OE EC cells, not Slug knockdown cells.
9. Discussion section is overlong with several redundancies and excessive speculation in several parts, making uneasy reading of the ms. An effort should be made to summarize several parts and provide more concise and less speculative message to the readers. In addition, the last paragraph of the Discussion contains several non-proved facts, as only the effect of SDF1a in Slug action (Notch pathway and angiogenesis) has been directly tested in the present work, but not the other stimuli mentioned in that paragraph.

Reviewer #2:

Remarks to the Author:

In the revised manuscript typo mistakes have been corrected and the previously missing abstract has been added. My major concerns have been addressed mostly by discussion (not by experimental evidence), and the following concerns (previous points 1, 7 and 8) remain:

1. Since this study focused on the function of Slug in endothelial cells, it is absolutely important to show that the phenotypes occur in endothelial-specific knockout mice rather than the global knock out mice used in this study. This cannot be replaced by in vitro studies using endothelial cells.
2. E-boxes of Dll4 promoter is bound by Snai1 and showed a regulatory role by a previous study and Slug as a family member of Snai1 potentially holds the same role. However, in order to draw a statement of "the direct suppression", a Dll4-promoter assay with Slug overexpression and knockdown is nevertheless needed.
3. Now the compensatory role of Snai1 during the absence of Slug has been discussed in the manuscript However, the rescue experiment (overexpression of Snai1 in Slug KO mice) would still be needed to make this point.

Reviewer #3:

Remarks to the Author:

The authors have done a careful review; their rebuttal is well under built and I have no further criticisms. The paper is improved and mature.

Response to Reviewers

We thank the reviewers for their second round of detailed and careful reviews. We address the comments of each reviewer individually and point-by-point. In this round of revision, we performed additional experiments to address the reviewers' concerns regarding the tumor knockout data. We also include clarification regarding the role of Slug as a potential transcriptional repressor of Dll4. Furthermore, we include additional discussion regarding the stated limitations of our study: 1) the lack of definitive proof of Slug as direct transcription repressor of its target genes; 2) the lack of direct and empirical evidence excluding a potential compensatory role for Snail in SlugKO animal vascular development. We have done our utmost to address the questions raised, and in aggregate our new data further strengthen the conclusion that Slug plays an important role in developmental angiogenesis and a critical role in pathological angiogenesis, and that it does so through regulation of Notch signaling.

Our detailed responses (in black text) to reviewer comments (in grey italic text) are provided below.

Reviewer #1 (Remarks to the Author):

This is an improved version of the original ms. by Hultgren et al. describing Slug involvement in the control of EndoMT and in angiogenesis. In the present revised version the authors have addressed most of the concerns of this reviewer by performing additional experiments and/or providing solid arguments. They also provide additional evidence for the participation of Slug in in vivo angiogenesis by performing elegant Slug-EC rescue experiments in Slug KO mice. While most of the previous queries have been properly answered there are still some concerns that have not been fully addressed requiring additional elaboration.

Besides, there are some additional points in the present revised version that require clarification before publication

We thank Reviewer #1 for their careful reading of our revised manuscript and we appreciate their earlier feedback that helped to improve the revised version. We now provided additional clarification to the remaining points raised by Reviewer #1 as shown below.

Overall appraisal of the revised version:

My previous concern on the lack of novelty of the present work based on the previous in vitro study of the authors (Welch-Reardon et al., 2014) and the previous work of other group reporting a similar role for Snail in in vivo angiogenesis (Wu et al., Nat Comms, 2014), has been properly addressed by the authors in the rebuttal letter and additional support from the revised ms. Similarly, the authors have provided stronger evidence and arguments to support the proposal that Slug mediates a partial EndoMT process in EC in vivo and in vitro. The doubts on the analyses of RNA-seq data are also much

clarified. I am now more convinced on the interest and relevance of the present study regarding Slug involvement in physiological angiogenesis, but its relevance for tumor angiogenesis is still not completely clear.

Main points:

Related to the original review:

1. Tumor data in Slug KO model.

I appreciate and thank the authors for the detailed information and clarification of the specific Slug KO model used here, and the differences with Slug KO models used in other studies that were not evident to this reviewer in the original ms. Also, I appreciate the clarification regarding the experimental tumor model of xenografting “wild type” CRC cells into either wild type or constitutive Slug KO mice. They have also included some of the requested information on the mouse CRC cell lines, including appropriate references (#32,33) (please, note that ref. 31 is not appropriate to this point).

We thank Reviewer #1 for pointing out the mistake in the reference number in the previous “Answer to Reviewer Comments”. The reference has now been corrected and the current reference numbers for each cell line used are #32 (CMT-93), #33 (MC-38) and #34 (current version, B16).

However, the authors still do not indicate the differences between the two CRC cell lines used, that are described in the original references: CMTP3 (epithelial and low tumorigenic cells) and MC38 (mesenchymal and highly tumorigenic cells) (#32 y 33) that can be important to consider here.

We agree that the difference in tumor grade in these two CRC lines is an important detail and could be critical for data interpretation, especially if they showed a difference in outcome in a SlugKO host background. However, with both lines, we observed a dramatic reduction in tumor growth regardless of differences in inherent tumorigenicity. This suggests that Slug expression in tumor stroma – and specifically in the vasculature, is the critical determinant of tumor size, and that absence of endothelial Slug in the SlugKO animal restricts tumor growth. We have now included this point in the Discussion (page 21, highlighted in second paragraph)

In response to a related question on the cell lines raised by reviewer #3 on endogenous Slug expression, the authors argue that “based on previous studies showing Slug expression in both CMT-93 and MC-38 tumor cells (Ref 31, 32), we would expect them to express Slug in our assays”. I would like to remark that Slug expression in refs 32, 33 (not 31 and 32) is analyzed by RT-PCR with apparent low or very low Slug mRNA levels in MC38 cells (33). I still think it is important to know whether tumor cells express or not endogenous Slug to correctly interpret the results of tumor xenografts and, thus, the authors should confirm Slug expression in the two CRC cell lines at protein level, by WB.

We have now included a Western blot showing the Slug expression level in each of the three tumor lines used in our study (Suppl. Fig.S2, and below). For expression controls, we used wild-type EC (i.e. quiescent, with low endogenous Slug) as well as SlugOE EC. Importantly, we found that not only is Slug expression intact, but also relatively similar, across all three cancer cell lines. Thus, variations in Slug expression in the tumor cells do not explain our finding that tumor size is reduced across all three cell lines in a SlugKO host background.

In addition, they need to explain the apparent contradictory data on tumor growth observed in MC38 cells at end points presented in Fig. 1B (no apparent tumor growth at end point) and Suppl Fig. S1a (left, median tumor growth around 500 mm³). Indeed, if the latter data are correct, MC38 tumors should be analyzed for angiogenesis and necrosis, instead of CMT93 tumors presented in Suppl Fig. S1b.

We thank Reviewer #1 for identifying this apparent inconsistency in our manuscript. The differences observed in the MC-38 tumor xenograft experiments reflect differences between experimental cohorts. Data from one round were presented in Fig. 1B, whereas the end-point tumor size graph (Suppl. Fig. S1a, left panel in the previous version, Suppl. Fig. S1b, bottom right panel, in the current version) was generated from a different cohort of animals. We apologize that we did not make the methodology sufficiently clear in the earlier presentation of these data and have now included the tumor growth and end-point tumor size graphs from each of these cohorts (Fig. 1B, Suppl. Fig. S1a, b). For each we saw a similar trend of reduced tumor growth in the SlugKO mice. Compared to the first cohort, more tumors grew in both WT and SlugKO mice in the second cohort. In each case, however, MC-38 cells were more variable in tumorigenicity than CMT-93 cells. While these nuances in our data are interesting and raise fascinating questions regarding the specific differences in tumorigenicity of CMT-93 vs. MC-38 cells in these studies, further investigation of this phenomenon is clearly beyond the scope of the current study. Our point remains that for both tumors, their ability to grow in a SlugKO mouse was severely restricted, and this correlated with a profound decrease in tumor angiogenesis. Because data from CMT-93 were more consistent across all experimental animals, we elected to focus additional studies on CMT-93 as our model cancer cell line.

Moreover, I still think that analyses of the proliferative status of the xenografted tumors in wild type and Slug KO mice is strictly required to complement the analysis of

performed in Wu et al. for Snail) should be included in the present ms. to unambiguously support the proposed repression action of Slug on Dll4 transcription, as well as on the Jag2 promoter. Otherwise, only Slug binding to those promoter regions can be claimed but not functional transcriptional effects.

The critical point here is the species being studied in each instance. The Wu et al paper studied mouse embryos since EC-specific Snail knockout is embryonic lethal, and they therefore described signaling in murine cells. They elegantly and definitively showed with ChIP and promoter assays in murine EC that Snail directly represses the murine Dll4 promoter. In contrast, in our paper, we complemented our *in vivo* mouse model with 3D angiogenesis assay using primary *human* endothelial cells. Specifically, for our ChIP experiments, we used HUVEC (human umbilical vein endothelial cell). This is why the E-box locations reported in the two manuscripts are different – Wu et al studied the murine gene while we studied the human gene (see figure below).

To address this potential confusion, in addition to our reference of the Wu et al paper (Ref 55 in the previous version, Ref 57 in the current version), we have also included an additional reference wherein Slug ChIP was performed on human cells (Ref 58). To further clarify, we now also include the exact primer locations (see figure below).

We agree with the reviewer that without the promoter assay, we cannot definitively prove functional effects of Slug binding to the promoter region of Dll4 and Jag2. For that reason, we have now changed our language to reflect this limitation of our study (Results, page 17, highlighted in top paragraph).

2. Knockdown analyses of CXCR4 and CXCR7 (Fig. 7H, S6h) lack proper controls to show efficient knockdown of the corresponding genes (i.e., decreased endogenous CXCR4 and CXCR7 mRNA levels).

Actually, these data were shown in the previous versions in Suppl. Fig. S6b and c. These data are now shown in Suppl. Fig. S13a (and below) in the current version.

3. IF images presented in Fig. S4g are of low quality (high background and apparent non-specific stain). Please, improve them or delete this specific information.

We have now removed this supplemental figure.

4. Western blots presented in Fig. 5C and S5a, lack lane labeling and loading controls (except for Dll4 expression).

We have now included lane labeling. We have furthermore clarified that because we re-probed for protein expression in the same or duplicate membranes, that the loading control shown for Dll4 applies across all of these panels.

5. In Fig. 1D, please include magnification areas of matrigel plugs of Slug KO mice to more faithfully compare with the plugs of WT mice.

We have now included magnification area for Matrigel plugs of SlugKO mice for comparison (Fig. 1D).

6. Data of lentiviral Slug overexpression in EC, shown in Fig. 3A and S3c, appear to this reviewer to be similar, or the same, as the data presented in the previous work of the group (Fig. 3 in Welch-Reardon et al., 2014). Please, clarify the differences.

The data shown in the current manuscript were all generated specifically for this manuscript. The experiment presented in the previous work (Fig.3 in Welch-Reardon et al., 2014) aimed to test whether Slug overexpression leads to preferential localization at the tip-cell position of developing sprouts. The experiment in the current manuscript

(Fig.3A and Suppl. Fig. S7) was designed to examine the differential effect of low and high Slug overexpression (as shown in Fig. 3, and in Suppl. Fig. S3 in the previous version and Fig.3 and Suppl. Fig. S7 in the current version) at the early (tip cell formation) and late stages (sprout extension and lumen formation) of angiogenesis. Thus, the experiment, while superficially similar, was to test out different ideas regarding Slug function. Some overlap in experimental outcome between the two studies also serves to validate assays over time and in different hands. We have revised the text to reflect these ideas better (Results, page 12, highlighted in first paragraph).

7. Page 11, 2nd paragraph: Conclusion on Slug role in developmental angiogenesis. Please, change “indispensable” for important.

We have now incorporated this suggestion (Results, page 11, highlighted in last line).

8. Page 16, 2nd paragraph, lines 3-6. Please note that information presented in Fig S5a refers to Slug OE EC cells, not Slug knockdown cells.

We thank the reviewer for pointing out this error. It is now corrected.

9. Discussion section is overlong with several redundancies and excessive speculation in several parts, making uneasy reading of the ms. An effort should be made to summarize several parts and provide more concise and less speculative message to the readers. In addition, the last paragraph of the Discussion contains several non-proved facts, as only the effect of SDF1a in Slug action (Notch pathway and angiogenesis) has been directly tested in the present work, but not the other stimuli mentioned in that paragraph.

We have now revised the discussion section to be more concise. We have also adjusted our language so that it is more clear which points were demonstrated by data in this study and which points are speculations and ideas for future directions.

Reviewer #2 (Remarks to the Author):

In the revised manuscript typo mistakes have been corrected and the previously missing abstract has been added. My major concerns have been addressed mostly by discussion (not by experimental evidence), and the following concerns (previous points 1, 7 and 8) remain:

We thank Reviewer #2 for noting improvements to the manuscript. We would dispute however, the degree to which new experiments have addressed the reviewer's concerns. We made strong efforts, whenever feasible, to add new experiments and data to address Reviewer #2's concerns (specifically points #1 and 8).

We have now made major structural changes to the figures, and especially the supplemental figures, for the sake of clarity. Hopefully these changes have improved the flow of the manuscript and made it easier to locate the experiments and the data. In addition, we hope our point-by-point answers below will further alleviate the reviewer's concerns.

1. Since this study focused on the function of Slug in endothelial cells, it is absolutely important to show that the phenotypes occur in endothelial-specific knockout mice rather than the global knock out mice used in this study. This cannot be replaced by in vitro studies using endothelial cells.

We believe we have addressed this concern as we do not rely on *in vitro* studies to make the case that EC-specific Slug is required for proper angiogenesis. In the previous revision we included an *in vivo*, EC-specific rescue experiment that we believe provides compelling support for our hypothesis. We did this by re-expressing Slug only in the developing mouse retinal vascular EC in otherwise global SlugKO mice (Fig. 2J, K, Suppl. Fig. S2k, l,m,n,o in the previous version, Fig. 2J, K, Suppl. Fig. S6 in the current version, or see below). This significantly restored retinal angiogenesis.

Specifically, we generated a lentiviral construct with a dsRed-STOP expression cassette flanked by 2 loxP sites, followed by either a GFP alone or a mouse Slug-GFP expression cassette (Suppl. Fig. S2k in the previous version, Suppl. Fig. S6a in the current version, or see below). We bred our global SlugKO mice into a VE-cad promoter driven Cre mouse background so that they are still SlugKO in all the cell types but express Cre recombinase in the EC where VE-cad is expressed. We produced high titer virus, in the range of 1.5×10^8 FU/mL or higher. We then performed retro-orbital injection of the virus to VEcad-Cre⁺, SlugKO mice at P6. The mice were sacrificed at P13 and their retinas were harvested, stained and imaged. VEcad-Cre⁺, SlugKO mice injected with GFP virus, VEcad-Cre⁻, SlugKO animals injected with mSlug-GFP virus and animals without injection were used as negative controls and to control for potential damage caused by the procedure. Using this method, we observed Slug re-expression in only the retinal EC in the VEcad-Cre⁺ animals injected with mSlug-GFP virus and we

observed an increase in vascularization index (area covered by vasculature) in these animals compared with negative controls. Thus this *in vivo* experiment further strengthens our hypothesis that Slug expression in the endothelial cells specifically drives and is important for vascular morphogenesis.

J

K

a

b

c

d

e

2. E-boxes of Dll4 promoter is bound by Snai1 and showed a regulatory role by a previous study and Slug as a family member of Snai1 potentially holds the same role. However, in order to draw a statement of “the direct suppression”, a Dll4-promoter assay with Slug overexpression and knockdown is nevertheless needed.

We agree with the reviewer that a promoter assay would definitively prove functional significance of Slug binding to the Dll4 promoter. Due to time constraints as well as the existing size of our manuscript, we have chosen not to include this study. We have therefore adjusted the text to “suggest” a functional link (Results, page 17, highlighted in top paragraph).

3. Now the compensatory role of Snai1 during the absence of Slug has been discussed in the manuscript. However, the rescue experiment (overexpression of Snai1 in Slug KO mice) would still be needed to make this point.

We believe the mouse has already done this experiment for us. We did not find an increase in Snai1 expression in FACS sorted retinal EC from the SlugKO mice (Suppl. Fig. S2j in the previous version, Suppl. Fig. S5f in the current version, and see below). Thus, a physiological level of Snai1 in the EC is not sufficient to compensate for the loss of Slug. While it is possible that overexpression of Snai1 might rescue, it would be hard to argue that this was a physiologically relevant result.

In addition, our previous study (Welch-Reardon et al. 2014) had demonstrated that knockout of *either* Slug or Snail leads to sprouting defects, indicative of a non-redundant role for these two transcription factors in vascular growth. Furthermore, our previous study also suggested that these two transcription factors were induced at different times during sprouting angiogenesis (Slug is induced earlier than Snai1. Welch-Reardon et al. 2014, Fig.1B & Fig.S1A), again suggesting different roles for each gene in the context of sprouting angiogenesis.

Slug expression in mouse retinal EC

Snail expression in mouse retinal EC

Reviewer #3 (Remarks to the Author):

The authors have done a careful review; their rebuttal is well under built and I have no further criticisms.

The paper is improved and mature.

We greatly appreciate Reviewer #3's response to our revisions and thank them for their contributions.

Reviewers' Comments:

Reviewer #1:

Remarks to the Author:

In this second revised version the authors have satisfactorily addressed most of the specific previous concerns.

There is only one specific comment on the Discussion section that in my opinion is still overlong and should be shortened it further.

A mistake is also found in page 9, 2nd line from bottom:

"...Slug KO retinal EC (Fig, 2I, S4a)", ref to Fig S4a should be "S5a".

Otherwise, the ms. is acceptable for publication

Reviewer #2:

Remarks to the Author:

This is a second time-revised manuscript by Hultgren et al. investigating the function of Slug protein in EndMT during angiogenesis. In the present revised manuscript, the authors have made structural changes to the figures. Now the flow of the manuscript has been improved, and the concept is easier to understand. Overall my comments have been satisfactorily addressed.

I still have one minor concern, though:

In the supplementary figure 6b, the same red fluorescence image seems to be used in two different experiments (Switch GFP and Switch mSlug-GFP transfection.)

REVIEWERS' COMMENTS:

Reviewer #1 (Remarks to the Author):

In this second revised version the authors have satisfactorily addressed most of the specific previous concerns.

There is only one specific comment on the Discussion section that in my opinion is still overlong and should be shortened it further.

We have now done our best to make the discussion more concise without missing any main points.

A mistake is also found in page 9, 2nd line from bottom:

"...Slug KO retinal EC (Fig, 2l, S4a)", ref to Fig S4a should be "S5a".

We apologize for the typo. It is now corrected.

Otherwise, the ms. is acceptable for publication

Reviewer #2 (Remarks to the Author):

This is a second time-revised manuscript by Hultgren et al. investigating the function of Slug protein in EndMT during angiogenesis. In the present revised manuscript, the authors have made structural changes to the figures. Now the flow of the manuscript has been improved, and the concept is easier to understand. Overall my comments have been satisfactorily addressed.

I still have one minor concern, though:

In the supplementary figure 6b, the same red fluorescence image seems to be used in two different experiments (Switch GFP and Switch mSlug-GFP transfection.)

We apologize for this image processing error. Correct image is now used for the Switch GFP transfection (Supplementary Fig. 6b).